# Hamster organotypic modeling of SARS-CoV-2 lung and brainstem infection

Marion Ferren [1✉], Valérie Favède[1,2], Didier Decimo[1], Mathieu Iampietro [1], Nicole A. P. Lieberman [3], Jean-Luc Weickert[4], Rodolphe Pelissier [1], Magalie Mazelier[1], Olivier Terrier [5], Anne Moscona [6,7,8,9], Matteo Porotto[6,7,10], Alexander L. Greninger[3], Nadia Messaddeq[4], Branka Horvat [1] & Cyrille Mathieu [1✉]

SARS-CoV-2 has caused a global pandemic of COVID-19 since its emergence in December 2019. The infection causes a severe acute respiratory syndrome and may also spread to central nervous system leading to neurological sequelae. We have developed and characterized two new organotypic cultures from hamster brainstem and lung tissues that offer a unique opportunity to study the early steps of viral infection and screening antivirals. These models are not dedicated to investigate how the virus reaches the brain. However, they allow validating the early tropism of the virus in the lungs and demonstrating that SARS-CoV-2 could infect the brainstem and the cerebellum, mainly by targeting granular neurons. Viral infection induces specific interferon and innate immune responses with patterns specific to each organ, along with cell death by apoptosis, necroptosis, and pyroptosis. Overall, our data illustrate the potential of rapid modeling of complex tissue-level interactions during infection by a newly emerged virus.

[1] CIRI, Centre International de Recherche en Infectiologie, Team Immunobiology of the Viral infections, Univ Lyon, Inserm, U1111, CNRS, UMR5308, Université Claude Bernard Lyon 1, Ecole Normale Supérieure de Lyon, LYON, France. [2] Département du Rhône, Lyon, France. [3] Department of Laboratory Medicine, University of Washington Medical Center, Seattle, WA, USA. [4] Institut de Génétique et Biologie Moléculaire et Cellulaire (IGBMC), INSERM U1258, CNRS UMR 7104, Université de Strasbourg, Illkirch, France. [5] CIRI, Centre International de Recherche en Infectiologie, Team VirPath, Univ Lyon, Inserm, U1111, CNRS, UMR5308, Université Claude Bernard Lyon 1, Ecole Normale Supérieure de Lyon, LYON, France. [6] Center for Host-Pathogen Interaction, Columbia University Medical Center, New York, USA. [7] Department of Pediatrics, Columbia University Medical Center, New York, USA. [8] Department of Microbiology & Immunology, Columbia University Medical Center, New York, USA. [9] Department of Physiology & Cellular Biophysics, Columbia University Medical Center, New York, USA. [10] Department of Experimental Medicine, University of Study of Campania 'Luigi Vanvitelli', Naples, Italy. ✉email: marion.ferren@inserm.fr; cyrille.mathieu@inserm.fr

In late 2019, the emergence of the severe acute respiratory syndrome coronavirus 2 (SARS-CoV-2) led to a global pandemic of COVID-19. As of August 2021, >215 million laboratory-confirmed cases were reported, and >4.5 million patients died worldwide from this disease[1]. SARS-CoV-2 infection induces severe acute respiratory syndrome, which can also be associated with central nervous system (CNS) infection and neurological symptoms including smell dysfunction, headache, muscle pain, myopathy, and in rare cases, generalized myoclonus, ischemic stroke, and perivascular acute disseminated encephalomyelitis[2–7]. It has been suggested that SARS-CoV-2 reaches the medulla oblongata and that brainstem infection may be involved in both respiratory and heart failure in patients[8–12]. To date, the neuro-invasive potential of SARS-CoV-2 in humans remains poorly understood[13,14]. The great majority of studies show that most of the symptoms reflecting CNS affection are related to brain blood vessel infections[15–22]. Although less common, the susceptibility of human neurons to the infection and the permissiveness of human brain organoids was observed in vitro[23–27] and SARS-CoV-2 viral particles or RNA have already been found in the cerebrospinal fluid[28] and in the brain of a subset of patients[12,29]. The low representation of SARS-CoV-2 in CSF may be related to a poor ability to bud in this tissue as observed for measles virus (MeV) or to any limitation to invade the brain in the most severe respiratory cases which represent the large majority of the samples. In order to reach the CNS, SARS-CoV-2 may travel from the periphery into the CNS through the olfactory neurons or through the vagus nerve from the lungs or gut. In addition, SARS-CoV-2 infection has been shown to disrupt the blood-endothelial barrier by damaging the choroid plexus epithelium and as a consequence of cytokine storm and systemic inflammation[27,30,31].

SARS-CoV-2 is an enveloped, positive-sense, single-stranded RNA virus that belongs to the *Betacoronavirus* genus within the *Coronaviridae* family. The infection starts with the attachment of the viral surface glycoprotein, Spike (S), to the human angiotensin-converting enzyme 2 (ACE2) at the surface of the target cell[32]. To execute its functions, the S must be in its protease-cleaved form composed of the S1 and S2 subunits. The activity of the cellular transmembrane serine protease TMPRSS2 highly correlates with viral dissemination, suggesting that it may participate in the processing of S, but S can also be cleaved by the endosomal proteases Cathepsin B and Cathepsin L[33]. In addition, cell entry of SARS-CoV-2 can be pre-activated by the proprotein convertase furin, reducing the virus's dependence on target cell proteases for entry[34,35]. The S1 fragment of the Spike protein cleaved by furin can also bind to Neuropilin-1, which is abundantly expressed at the surface of endothelial and epithelial cells. This binding may also facilitate SARS-CoV-2 infection by promoting the viral interaction with ACE2[36,37].

Most observations of the pathogenesis of SARS-CoV-2 arise either from in vitro studies or post mortem analysis of infected patients, and there is a need for models that can help decipher the initial steps of infection in real time. To date, different animal models have been described to study SARS-CoV-2 pathogenesis, transmission, or antiviral efficacy, including transgenic mice expressing human ACE2, hamsters, ferrets, rhesus macaques, cynomolgus macaques, and African green monkeys[38].

Golden Syrian hamsters have been shown to be a relevant small animal model for several viruses[39] and more specifically for respiratory viruses that also target the CNS, such as the paramyxoviruses Nipah virus (NiV)[40] and MeV[41]. The pathogenesis of SARS-CoV[42,43] and SARS-CoV-2[44–47] in Syrian hamsters is similar to that observed in humans, supporting the use of hamsters as models for studying these infection[45,48,49]. Cerebellar and hippocampal organotypic cultures from small rodent models have been characterized in the laboratory and shown to be relevant for studying CNS infection by neurotropic viruses and screening antiviral drugs[50]. These ex vivo cultures offer a unique opportunity to access the infected organ directly in order to observe early viral tropism. In this study, we characterize two new three-dimensional (3D) organotypic culture models obtained from suckling hamster brainstems and lungs and compare SARS-CoV-2 infection with that of two neuro-invasive respiratory viruses (NiV and MeV) in lung and brainstem ex vivo models.

Here, we show that both organotypic systems retain their relevant physiological properties for the duration of the experiment (4 days) and are susceptible to NiV and SARS-CoV-2 infection, whereas a hyperfusogenic encephalitic MeV strain infects only the brain organotypic cultures. Type 1 and type 2 pneumocytes, as well as ciliated cells, are susceptible to infection in hamster lung explant cultures, reflecting what has been shown in humans[51]. As for the hamster brainstem, we show that SARS-CoV-2 targets granular neurons. Real-time characterization also highlights the induction of type I and III innate immune responses and an inflammatory response to infection at the organ level, as well as the caspase-3-independent apoptosis, necroptosis, and pyroptosis cell death signatures, during the first days of SARS-CoV-2 infection in both organs. Taken together, our results pave the way for the use of these models to study SARS-CoV-2 infection in organs and assess the efficacy of candidate antivirals before in vivo validation.

## Results

**Hamster lung and brainstem ex vivo cultures are viable and susceptible to SARS-CoV-2 infection.** Since SARS-CoV-2 notably targets the upper respiratory tract and lungs and may also infect the brainstem, we have developed new ex vivo models of these organs from naive suckling hamsters, based on our previous experience using organotypic cerebellum cultures[50]. Lungs and brainstems were isolated and sliced, at 500 μm and 350 μm thicknesses, respectively, based on the stability of each structure in the slicing process. The 3D cultures were then maintained on a polytetrafluoroethylene (PTFE) membrane in order to keep an air–liquid interface for up to 4 days (Fig. 1a). In opposition with primary cultures, organotypic cultures are usually not fully soaked in the medium in order to allow oxygenation[52]. The 0.4 μm semipermeable pores of the insert permit the diffusion of the medium into the cultures. Metabolic activity, the main parameter reflecting the viability of the cultures, did not decrease over time as quantified by Alamar blue assay (Fig. 1b). To evaluate the susceptibility of ex vivo cultures to SARS-CoV2 infection, we first quantified the activity of its entry receptor ACE2. Then, we quantified the expression level of the proteases known to cleave and mature the viral surface Spike glycoprotein (S) and the viral receptors ACE2 and Neuropilin-1. The ACE2 activity/μg of total protein lysate from hamster organs on the day of slice preparation was assessed in the lung and brainstem cultures and compared with cerebellum cultures used as a reference. An ACE2 inhibitor was used to confirm that the quantified activity was real. The three cultures were found to exhibit a similar ACE2 activity, with the lung displaying a slightly higher activity (Fig. 1c). The transcription level of *ACE2* and *Neuropilin-1* were quantified in non-infected cultures by RT-qPCR (Fig. 1d). This confirmed the mRNA expression of *ACE2* in both cultures and the higher *Neuropilin-1* mRNA expression in the lung cultures compared to brain cultures. The mRNA of *TMPRSS2*, *Cathepsin* B, and *Cathepsin* L, the three proteases that are known to cleave and activate SARS-CoV-2 S, was also quantified by RT-qPCR. As in human tissues, *TMPRSS2* was highly expressed in hamster lungs but below the quantification limit in the brainstem and

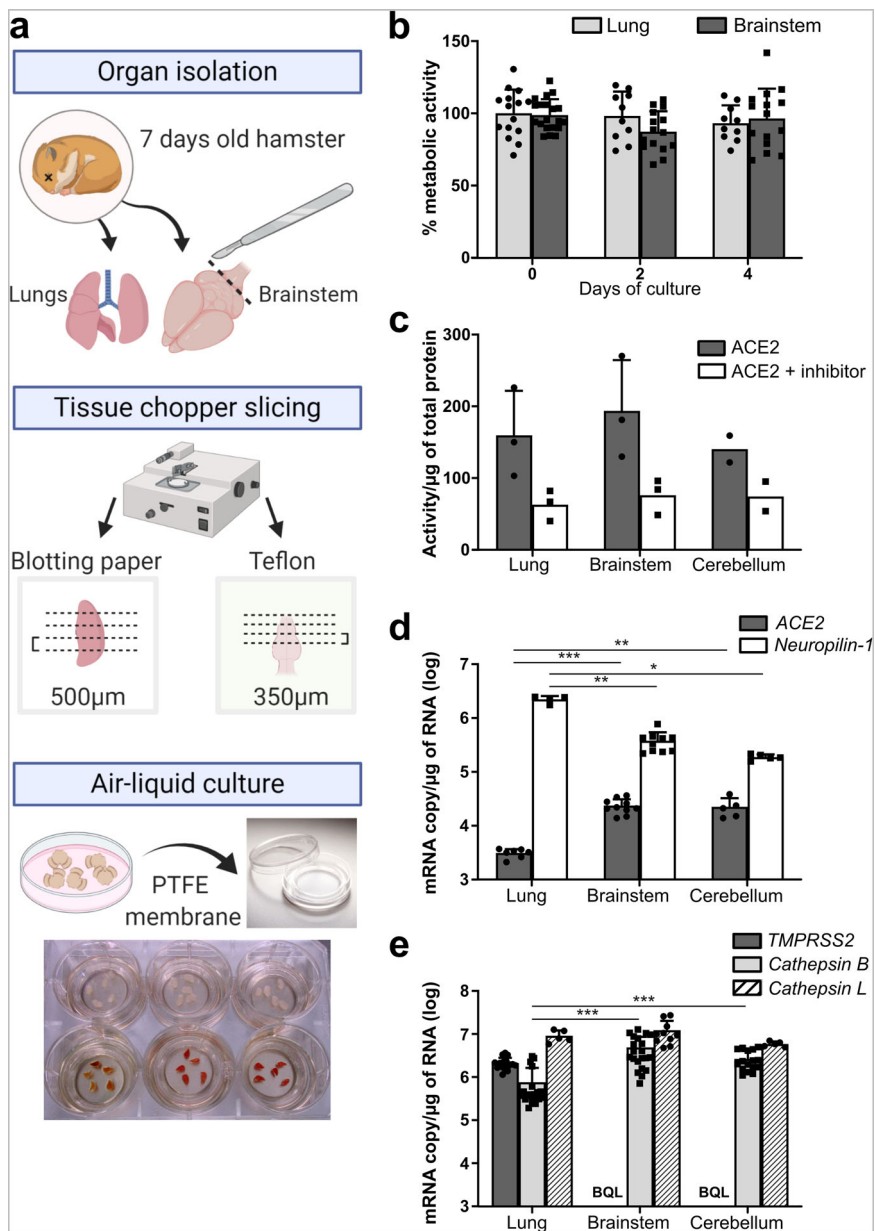

**Fig. 1 Characterization of the lung and brainstem organotypic cultures. a** Schematic representation of the generation of hamster organotypic cultures. **b** Cellular metabolism activity over time in % of day 0 of culture, quantified by the Alamar blue assay. ($n$ = minimum 10 biologically independent animals). **c** ACE2 activity quantified with the fluorometric ACE2 Activity Assay Kit. ($n$ = 3 biologically independent animals). **d** ACE2 and Neuropilin-1 basal mRNA expression ($n$ = minimum five biologically independent animals) and **e**TMPRSS2, Cathepsin B, and Cathepsin L basal mRNA expression in the models quantified by RT-qPCR (day 0 of culture). ($n$ = minimum 12 biologically independent animals). BQL below the quantification limit, ACE2 angiotensin-converting enzyme 2, PTFE polytetrafluoroethylene. Error bars represent SD. Statistical analyses were performed using the Kruskal–Wallis test. *$P < 0.05$; **$P < 0.01$; ***$P < 0.001$ Source data are provided as a Source Data file.

cerebellum[26,53]. On the contrary, Cathepsin B mRNA expression levels were significantly higher ($\approx 2*10^6$ mRNA copies/µg of RNA) in the brain cultures compared with the lung ($3.6 \times 10^5$ mRNA copies/µg of RNA) (Fig. 1e). Resources available to work with hamsters remain very limited, notably the antibodies for immunostainings. However, in order to look deeper into the expression level of ACE2, Neuropilin-1, and the serine proteases in the brain, all four cell populations from suckling hamster cerebella were sorted by flow cytometry (Supplementary Fig. 1f). Apart from TMPRSS2 that remained undetected, mRNA from ACE2, Neuropilin-1, and Cathepsin B and L were all expressed in neurons, oligodendrocytes, astrocytes, and microglia. mRNA expression of ACE2 was still below 1 copy per cell in all cell types,

even in neurons ($3.56 \times 10^2$ mRNA copies/10,000 cells), suggesting that few neural cells or their subsets express the receptor.

These results suggest that the lung organotypic models harbor all the main components required for SARS-CoV-2 infection and virus fusion at the membrane or in the endosome, whereas the brain slices might allow SARS-CoV-2 infection only via the endosomal pathway in a small subset of cells as implied by the lack of TMPRSS2 mRNA expression.

**Organotypic cultures are susceptible and permissive to SARS-CoV-2 infection.** In order to evaluate the infectability of our new models, hamster lung, brainstem, and cerebellum organotypic cultures were infected with recombinant SARS-CoV-2_neon

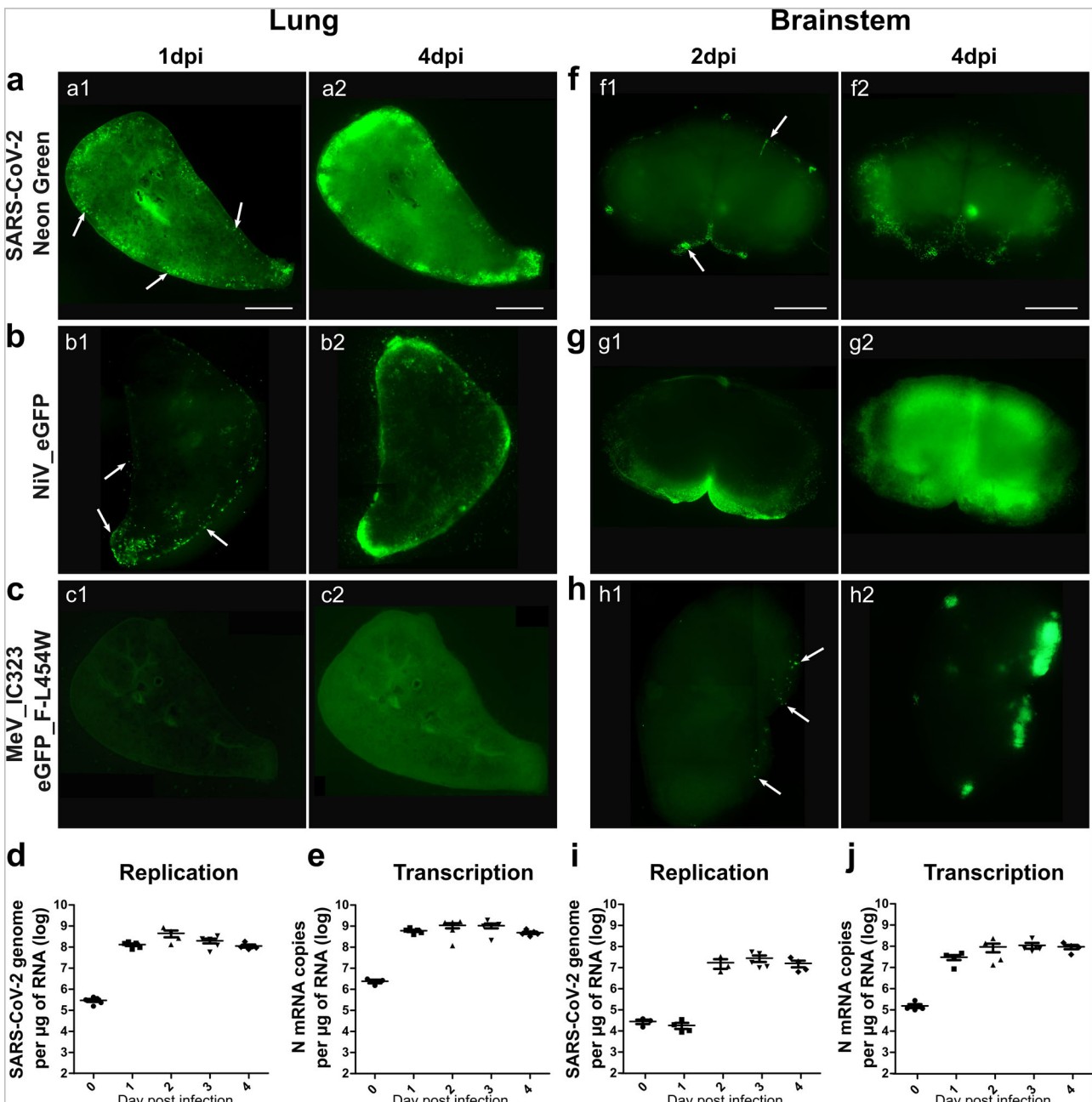

**Fig. 2 Hamster organotypic culture infection by three respiratory viruses and the dissemination of SARS-CoV-2. a–c, f–h** The entry of three different encephalitogenic respiratory viruses, icSARS-CoV-2-mNG (infection: 10,000 plaque-forming unit (pfu)), NiV-EGFP (infection: 5000 pfu) and the hyperfusogenic variant MeV IC323-EGFP-F L454W (infection: 1000 pfu) was monitored by following the fluorescence at 1 dpi (**a1**; **b1**; **c1**), or 2 dpi (**f1**; **g1**; **h1**) and 4 dpi (**a2**; **b2**; **c2**; **f2**; **g2**; **h2**). Pictures were taken using a Nikon Eclipse Ts2R microscope (500 ms of exposure), reconstituted using the Stitching plug-in with ImageJ software[119] and are representative of three independent experiments. Scale bar = 1 mm. **d, i** SARS-CoV-2 genomes per µg of total RNA and **e, j** SARS-CoV-2 N mRNA copies per µg of total RNA were quantified by RT-qPCR in the lung, and brainstem organotypic cultures at 90 min post infection and 1–4 days post infection (dpi) with 5000 pfu and normalized to the standard deviation for *GAPDH* mRNA. (*n* = 5 biologically independent animals). Source data are provided as a Source Data file.

green (icSARS-CoV-2-mNG)[54]. To compare the permissiveness of the organotypic cultures to other encephalitic respiratory viruses, they were infected in parallel with recombinant NiV and MeV, both expressing enhanced green fluorescent protein (EGFP) (referred to as rNiV-EGFP and MeV IC323-EGFP-F L454W, a CNS-adapted MeV that can infect in the absence of known receptors[55], respectively). Viral entry and dissemination were tracked by microscopy at 1 or 2 and 4 days post infection (dpi) (Fig. 2; Supplementary Fig. 2). Interestingly, icSARS-CoV-2-

mNG and NiV-EGFP entered and disseminated in both lung and brain cultures (Fig. 2a, b and Supplementary Fig. 2a, b), whereas MeV IC323-EGFP-F L454W only infected the brainstem and the cerebellum (Fig. 2c and Supplementary Fig. 2c). Even at 4 dpi the hyperfusogenic MeV did not infect any cells in the lung cultures, suggesting that MeV entry in lung epithelial cells may require additional factors in this model (Fig. 2c). The pictures taken 2 h following infection of the hamster organotypic cultures with 10,000 PFU do not show more fluorescence than the basal

autofluorescence observed in non-infected cultures (Supplementary Fig. 3a, b). On the contrary, neon green specific fluorescent cells are clearly detected at 4 dpi in hamster cultures infected with only 100 plaque-forming unit (PFU) of icSARS-CoV-2-mNG, confirming that the susceptibility of the brain cultures is not related to the high viral charge used for infection (i.e., 100 PFU/$10^7$ cells at least) (Supplementary Fig. 3c). All positive cells for neon green fluorescence were also positive for the immunofluorescent staining of SARS-CoV-2 nucleoprotein, confirming that the neon green fluorescence provides evidence of infection (Supplementary Fig. 3d). In order to test the relevance of our models and confirm that SARS-CoV-2 entry is specific regardless of the dose of virus, organotypic cultures from wild-type C57BL/6-suckling mice, which are supposed to be resistant to SARS-CoV-2 infection, were infected with 10,000 PFU of icSARS-CoV-2-mNG. The absence of fluorescence even at 4 dpi confirms that the dose of 10,000 PFU does not force viral entry in a non-susceptible model (Supplementary Fig. 3e, f). SARS-CoV-2 replication and transcription were then followed, respectively, by specific genome and nucleoprotein mRNA copy number quantification by RT-qPCR (Supplementary Fig. 6). The viral genomes and nucleoprotein transcripts were higher in the lungs than in the brainstems and cerebella (Fig. 2d, e, i, j; Supplementary Fig. 2d). These differences were already observed on the day of infection (Fig. 2d, e). The greater susceptibility of lung cultures might emerge either from the greater thickness of lung slices compared with that of brainstem and cerebellum cultures, or perhaps from a greater number of *Neuropilin-1*-expressing cells (Fig. 1d). In the lung, viral replication was very fast and the number of SARS-CoV-2 genomes/µg of RNA almost reached the plateau after 1 dpi ($1.2 \times 10^8$ genomes/µg of RNA) (Fig. 2d, e). In the brainstem and the cerebellum, transcription was initiated at 1 dpi. However, replication was delayed, with an increase in genome copies being observed only after 2 days, and by the end of the experiment reached lower values than detected in the lungs (Fig. 2i, j). The mRNA expression level of SARS-CoV-2 entry receptors and the proteases involved in the cleavage of the S was followed in the context of infection (Supplementary Fig. 1a–e). *ACE2* mRNA expression increased from days 3 and 4 post infection, but the *Neuropilin-1* mRNA expression did not vary over time (Supplementary Fig. 1a, b). Expression of *Cathepsin* B and L mRNA increased through time in both organs while *TMPRSS2* mRNA expression remained below the quantification limit in brainstem cultures (Supplementary Fig. 1c–e).

Altogether, these results show that all analyzed ex vivo cultures are susceptible to SARS-CoV-2 infection, although with slightly different kinetics.

**SARS-CoV-2 infection is blocked by remdesivir in organotypic lung and brainstem cultures**. Remdesivir showed antiviral efficacy in vitro and in vivo, and is in clinical use for COVID-19 treatment in certain countries[56–60]. We used it to validate our models for drug evaluation. The slices were treated daily at two different concentrations of remdesivir, one right over the IC90 in cell culture and one five times higher, as our organotypic cultures are more complex than regular monolayer cultures. The treatment started two hours after infection (Fig. 3a) and continued for up to 4 dpi to make sure that the treatment not only delays the infection but also blocks viral dissemination. Mock organotypic cultures were maintained under similar conditions and treated using a vehicle. After 4 days of treatment, the total amount of RNA extracted remained unchanged and the metabolic activity of both lung and brainstem slices remained very close to 100% and to that of non-treated slices, suggesting the very low, not to say null, effect of the drug on the metabolic activity at these doses

(Fig. 3b, c). The lower dose (2 µM) of remdesivir did not have a significant inhibitory effect on infection as assessed by RT-qPCR. However, after treatment with 10 µM of remdesivir, infection was reduced by almost 100% at 4 dpi in both lungs and brainstems.

Hydroxychloroquine has been shown to inhibit SARS-CoV-2 entry in vitro by acting on the endosomal pathway[56,61,62]. In order to evaluate how drug screening in organotypic cultures is predictive of in vivo results, we included hydroxychloroquine in our study. Based on the absence of *TMPRSS2* in brainstem cultures, we first speculated that viral entry in these cultures should occur through the endosomal pathway, where hydroxychloroquine should be effective. At the dose of 10 µM, hydroxychloroquine blocked >90% of the infection in our brainstem cultures (Fig. 3d). At 20 µM, the efficacy decreased concomitantly to the appearance of signs of toxicity highlighted by a 20% reduction of the total RNA extracted per culture compared with the non-treated samples (Fig. 3e). In the lung cultures, where both *TMPRSS2* and *Cathepsin B/L* are expressed, providing the virus with the ability to fuse both at the cell surface and in endosomes, hydroxychloroquine did not inhibit the infection significantly (Fig. 3d, e). Taken together, these data confirm that our model can be used to predict the in vivo efficacy of a drug.

To go further, we have tested totally different antiviral compounds such as Hsp90 inhibitors that also hold promise as modulators of SARS-CoV-2 infection. Derivatives of geldanamycin such as 17-DMAG have been developed to limit hepatotoxicity and are very promising in vitro studies but toxic in vivo[63–65]. Here, 17-DMAG treatment inhibits 28% of the infection in the lung and 90% in brainstem cultures (Fig. 3f). However, four times less of total RNA has been extracted from the treated cultures compared to the non-treated ones in both lung and brainstem (Fig. 3g). Our results confirm that organotypic culture models mimic precisely the patterns observed in vivo, and therefore, can be used for assessing drugs prior to in vivo experiments.

**SARS-CoV-2 preferentially targets neurons in the brain, and ciliated cells, type 1 and type 2 pneumocytes in the lungs**. To determine which cells are the main targets of SARS-CoV-2, viral tropism was evaluated by transmission electron microscopy (TEM) in lung and brainstem organotypic cultures, followed by immunofluorescent staining analyzed by confocal microscopy (Fig. 4). Based on the kinetics of viral replication, all the organotypic cultures shown in Figs. 4–6 were collected at 1 dpi for the lung and at 2 dpi for the brainstem and cerebellum. In lung cultures, infection was observed in type 1 and type 2 pneumocytes, as well as in the ciliated cells from the general area of the bronchioles (Fig. 4a–c). Immunofluorescence analysis confirmed these observations by showing the presence of SARS-CoV-2_S staining in cells positive for surfactant protein C (SP-C), Aquaporin 5 (AQP5), and α acetylated Tubulin (Tub) staining that are specific for type 2, type 1 pneumocytes, and ciliated cells, respectively (Fig. 4d–f). Most of the cells display microvilli as expected in young animals (Fig. 4c and Supplementary Fig. 4a). Infected cells harbored a large number of vacuoles and showed multiple signs of cell degradation: cytoplasmic material degradation, membrane coiling (blue star), large empty vacuoles (green arrow) (Fig. 4b1, b2). We observed autophagosomal vacuoles containing virions or degraded viral particles in all types of infected cells (Fig. 4a–c; Fig. 5a and Supplementary Fig. 4c). Virions were also found attached to the microvilli outside the cells (Supplementary Fig. 4a), and several cells showed disorganization of the smooth endoplasmic reticulum (SER), as well as accumulation of lipids and mitochondria that were undergoing degradation (Supplementary Fig. 4b).

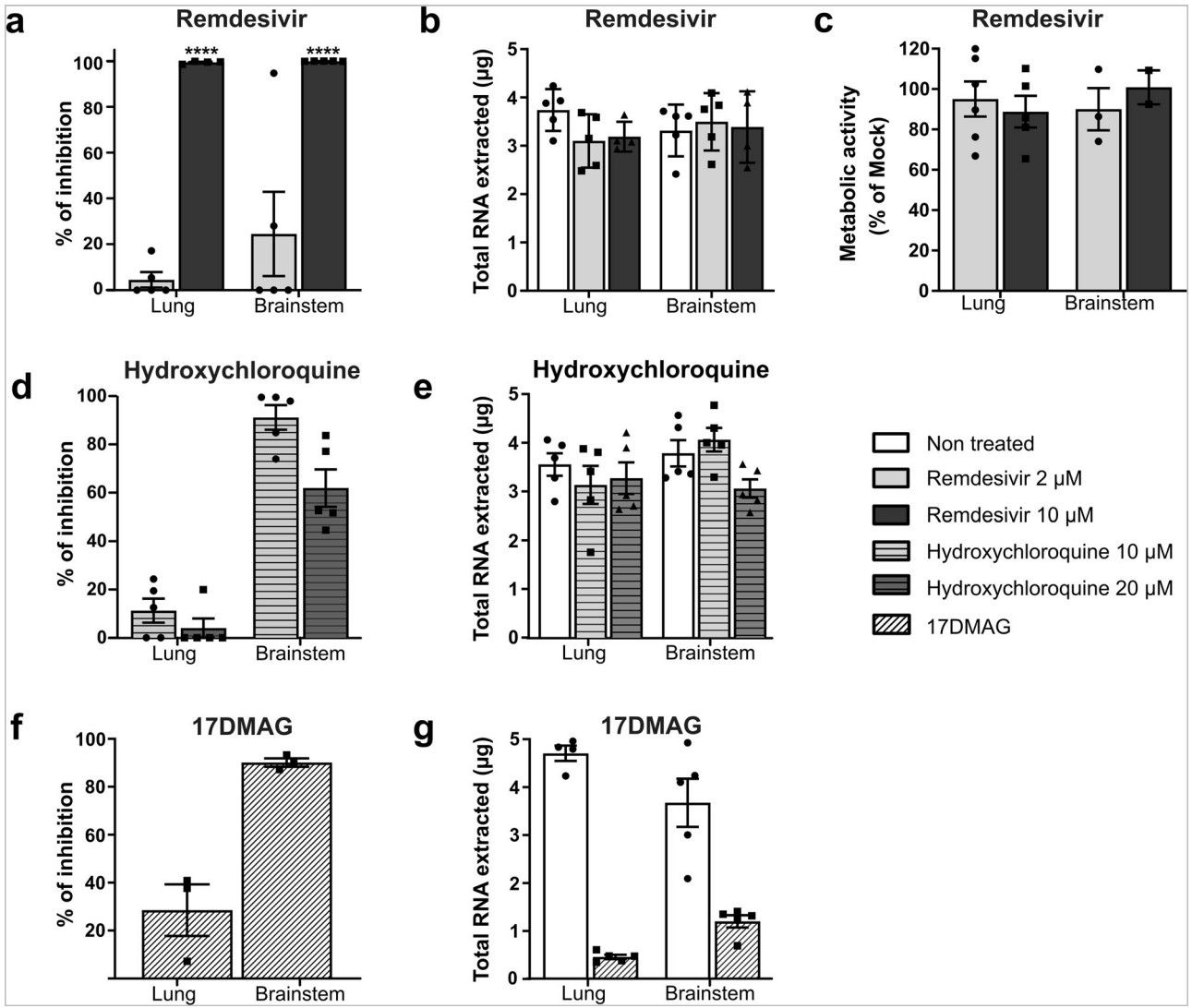

**Fig. 3 Antiviral activity of remdesivir, hydroxychloroquine, and 17-DMAG in hamster organotypic cultures infected by SARS-CoV-2. a, d, f** Organotypic cultures from hamsters were infected with SARS-CoV-2 at 1000 pfu/slice and treated at the indicated concentrations of remdesivir, hydroxychloroquine, and 17-DMAG at 90 min, 24, 48, and 72 h after infection ($n = 5$ biologically independent animals). Total RNA was harvested at 4 days post infection and the number of SARS-CoV-2 genomes was quantified by RT-qPCR. Results are expressed in % of inhibition of the infection compared with non-treated cultures. Statistical analyses were performed using the Kruskal–Wallis test. $*P < 0.05$; $**P < 0.01$; $***P < 0.001$. **b, e, g** Total RNA extracted per organotypic culture was quantified. **c** The toxicity of remdesivir on uninfected cultures that were treated in the same way was assessed via the Alamar blue assay. Results are expressed as the percentage of metabolic activity after 4 days compared with the non-treated samples. All error bars represent SD. Source data are provided as a Source Data file.

In the brainstem and in the cerebellum, TEM analysis revealed viruses in granular neurons (Fig. 5a and Supplementary Fig. 4d) with a developed Golgi apparatus. Moreover, the viral particles were often localized in double-membraned vacuoles (red arrows) inside the cells where other Coronaviruses are generally observed during their cell cycle (Fig. 5a1, a2)[66,67]. In the immunofluorescence analysis, SARS-CoV-2_S staining colocalized with NeuN-positive cells (Fig. 5b), confirming the infection of granular neurons. Myelin Basic Protein (MBP) staining surrounded the SARS-CoV-2_S without colocalization, suggesting that the cells positive for the infection could also be myelinated neurons and not oligodendrocytes (Fig. 5c). The cultures were also stained for microglia marker (Iba1), astrocytes (GFAP), and Olig2 (used as a second marker for oligodendrocytes), and SARS-CoV-2_S staining was not found in these cells (Fig. 5c–e). In the cerebellum, TEM analysis showed the infection of Golgi neurons with viral particles in autophagosomes. We did not observe

infection of Purkinje neurons (Supplementary Fig. 4d1, d2), suggesting that under the given conditions a selective infection of specific neuronal subtypes has occurred.

In these models, SARS-CoV-2 infects almost all epithelial lung cells but is selective for neuronal subtypes in the CNS. In both lung and brain organotypic cultures, the infection led to a marked cell degeneration that could conceivably affect organ functions.

**Apoptotic, necroptotic, and pyroptotic signatures are detected in both organotypic cultures**. Since unbalanced inflammatory responses can provoke organ failure, we evaluated cell death signatures in SARS-CoV-2-infected organotypic cultures. First, we performed TEM analysis, which highlighted the presence of apoptotic and necrotic cellular disorders in both infected lungs and brainstems as opposed to the non-infected cultures (Fig. 6a, b; Supplementary Fig. 5). The involvement of apoptosis was verified by transferase dUTP nick end labeling (TUNEL)

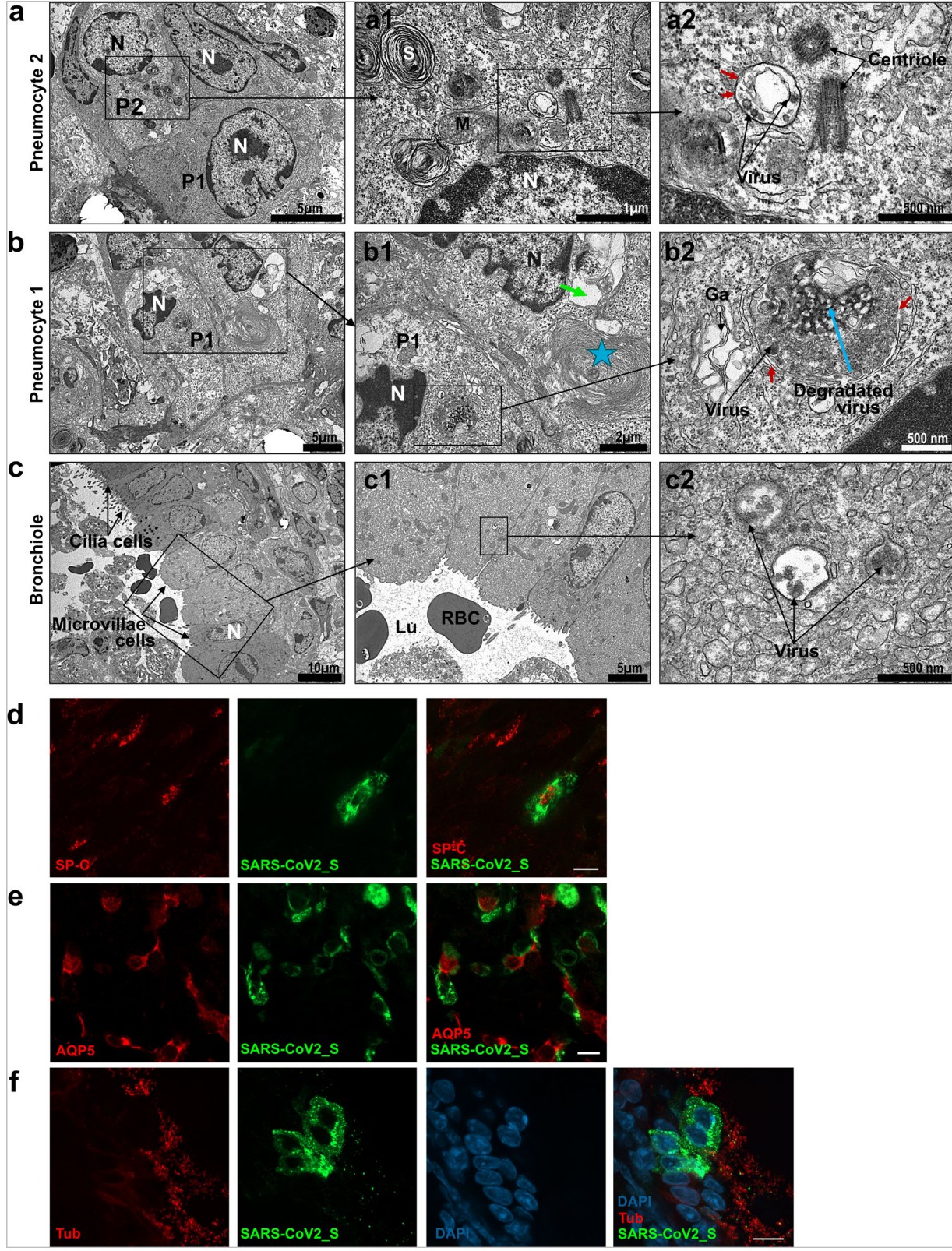

assays in both organotypic cultures (Fig. 6c, d). TUNEL staining was observed similarly in both non-infected and infected cultures, potentially as an artifact of the experimental procedure. However, most of the cells positive for SARS-CoV-2 staining were not positive for TUNEL, confirming that apoptotic cell death observed by TEM might not be a direct consequence of viral infection. Alternatively, cell death related to viral infection may

have been caspase-independent (Fig. 6c, d). Using RT-qPCR, we corroborated necroptotic events in line with microscopic observations despite an erratic expression of Tumor Necrosis Factor α (*TNFα*) throughout the four days of infection (Fig. 6f). As described in the former experiment (Fig. 6a, b), mRNA levels of the infected cultures were compared with uninfected cultures maintained under similar conditions ($n = 5$). Indeed, in both

**Fig. 4 SARS-CoV-2 tropism in hamster lung organotypic cultures during the first day of infection.** Cultures were infected with 1000 pfu of SARS-CoV-2 and fixed at day 1 post infection. **a–c** Ultrastructure of infected lung cells by transmission electron microscopy (TEM), scale bar is represented bottom right on each picture, representative of two independent experiments. **a** Low magnification of lung cells; P1 = type 1 pneumocyte, P2 = type 2 pneumocyte, N = nucleus. **a1** Enlargement of an infected type 2 pneumocyte with dense lamellar bodies in the cytoplasm (S = surfactant synthesis), a part of the nucleus, the centriole, mitochondria, and autophagosomal vacuoles containing virions. **a2** High magnification showing the centriole and the autophagosomal vacuole containing virions. Black arrows point to the double membrane. **b** Lung cells undergoing degeneration, displaying vacuoles and degraded cytoplasmic material. N = nucleus, P1 = type 1 pneumocyte. **b1** Enlargement of cells from the **b**, P1 cell contains an autophagosome containing virions, several vacuoles, and heterochromatin in the nucleus. The second cell exhibits membrane coiling (star) and large empty vacuoles (green arrow), indicating degradation. **b2** High magnification showing the double membrane of the autophagosome (red arrows) containing an accumulation of viral material (blue arrow). The black arrow show viruses surrounding the vacuole. Ga = swelled Golgi apparatus. **c** Respiratory bronchiole showing ciliated cells and microvillous cells. N = nucleus. **c1** Enlargement of the cells from **c**, showing microvillous cells and red blood cells (RBC) in the lumen of the bronchia (Lu). C2: high magnification showing three autophagosomes containing virions (arrows). **d–f** Lung cultures were stained with antibodies: anti-SARS-CoV-2_S, **d** anti-surfactant protein c (SP-C), **e** anti-Aquaporin 5 (AQP5), and **f** anti-α acetylated Tubulin (Tub). The immunofluorescence staining analysis was performed by confocal microscopy and is representative of three independent experiments. Scale bar 10 μm.

cultures, we observed a sharp increase in Mixed Lineage Kinase Domain Like Pseudokinase (*MLKL*) mRNA levels (Fig. 6e), which is known to be associated with Caspase 8 deficiency and Inflammatory Bowel Disease commonly observed in patients. Moreover, we showed that pyroptosis also occurs during viral infection, as inferred from the increase in *Gasdermin D* mRNA levels in both infected lung and brainstem cultures (Fig. 6g). Gasdermin D is also known to be a substrate of inflammation-related caspases, thus triggering an unbalanced inflammatory response potentially leading to organ failure[68,69]. Interestingly, although *Gasdermin D* levels decreased at day 4 post infection in the lungs, its expression kept increasing in the brainstem, possibly due to a difference in infection kinetics between the tissues. Furthermore, while the levels of Interleukin 18 (*IL-18*) mRNA remained low in both organotypic cultures (Fig. 6h), we documented a difference in the expression of Interleukin 1 β (*IL-1β*) mRNA that increased in the brainstem but remained low in the lungs (Fig. 6i). These data reveal that distinct cellular mechanisms leading to pyroptosis in the two organs.

**Innate and inflammatory responses are increased in both organotypic cultures.** To characterize the recapitulation of the responses of these models to SARS-CoV-2 infection, we transcriptomically profiled infected and uninfected organotypic cultures of both hamster brainstem and lungs (Fig. 7a–d). All results are presented here as the fold change of the infected condition compared to the non-infected one (*n* = 5). The transcriptomic first level of analysis pointed out the strong stimulation of the immune response with 19 and 20 out of 20 mainly altered Gene Ontology (GO) categories related to immunity in lung and brainstem, respectively (Fig. 7a, b). Alternatively, eight GO categories related to lymphocyte responses were altered in the brainstem versus two in the lung. To go further, the dichotomy in cellular responses occurring in both tissues at day 4 post infection was confirmed by the gene expression patterns, highlighting organ-dependent specificities in the host response to the infection (Fig. 7c, d). Indeed, the most significantly differentially expressed genes (DEG) in the lung include a plethora of upregulated interferon-stimulated genes (ISGs, Fig. 7c). Conversely, in the brainstem, these significant DEG contained many downregulated neuronal markers (Fig. 5d). This is consistent with the observation that the percentage of polyadenylated transcripts aligning to the SARS-CoV-2 genome is 6.74-fold higher in the lung than in the brainstem (2.90% vs 0.43%) (Fig. 7c, d).

In parallel, specific immunological markers including Myxovirus Resistance 1 (*MX1*), Interferon Stimulated Exonuclease Gene 20 (*ISG20*), C-X-C Motif Chemokine Ligand 10 (*CXCL10*), and C-C Motif Chemokine Ligand 5 (*CCL5*) mRNA were quantified by RT-qPCR and presented in terms of the fold change

between the infected and uninfected conditions (Fig. 7e–h). However, specific genes encoding ISG or chemokines were similarly upregulated in both organotypic cultures, pointing to these as potentially relevant to the innate immune response. Indeed, RT-qPCR profiles showed that the expression of *MX1* and *ISG20* ISGs and of *CCL5* and *CXCL10* chemokines were increased within the 4 days kinetics following SARS-CoV-2 infection in both cultures compared with the respective uninfected conditions. Interestingly, *ISG20* mRNA amounts decreased rapidly to lower levels, suggesting that its reduced antiviral exonuclease activity may be compensated at later times (Fig. 7f). Moreover, we noticed that, while exhibiting a similar trend, all these responses were delayed in brain cultures compared with the lungs (Fig. 7e–h).

## Discussion

SARS-CoV-2 infection starts in the lungs, inducing the severe acute respiratory syndrome regularly associated with neurological symptoms. Although CNS involvement has been a topic of extensive study, little attention has been paid to the possible role of brainstem infection in organ failure. In the ventral medulla oblongata, the preBötzinger complex is a defined neural network that is critical for generating the respiratory rhythm[70]. It is plausible that brainstem infection during its acute phase could affect both respiratory and cardiac function[8–12].

Organotypic cultures offer an opportunity to follow early steps of infection in real-time in a native 3D multicellular context[71]. Non-standardized lung ex vivo slices have been used for studying respiratory virus pathogenesis in multiple host species (from mice to monkeys)[72,73]. The highly standardized lung organotypic cultures we describe can be extended to several small animal models. Organotypic brainstem cultures have been previously characterized using 3–18-day-old mice or rats, with a focus on brain development, neuronal respiratory networks, and neurodegenerative diseases[74–77]. Our study provides a characterization of these models for investigating the initial 4 days of infection for the first time. From a single animal, we can prepare a large number of organotypic cultures (e.g., ≈8 from the brainstem, ≈20 from the lungs, and up to ≈10 from the cerebellum), allowing for numerous comparisons to be carried out in real time and simultaneously on several organs. In terms of assessing antiviral drugs prior to in vivo work, organotypic cultures are complementary to human airway epithelia (HAE) that are used for preliminary screening[56,78], since they are a more complex organ-like system. As such, they complement the HAE system, also a 3D lung model grown at an air–liquid interface, that has been recently used to evaluate SARS-CoV-2 antiviral molecules[79].

The use of viruses encoding a fluorescent protein allows the monitoring of the infection in real-time within tissues. The lung

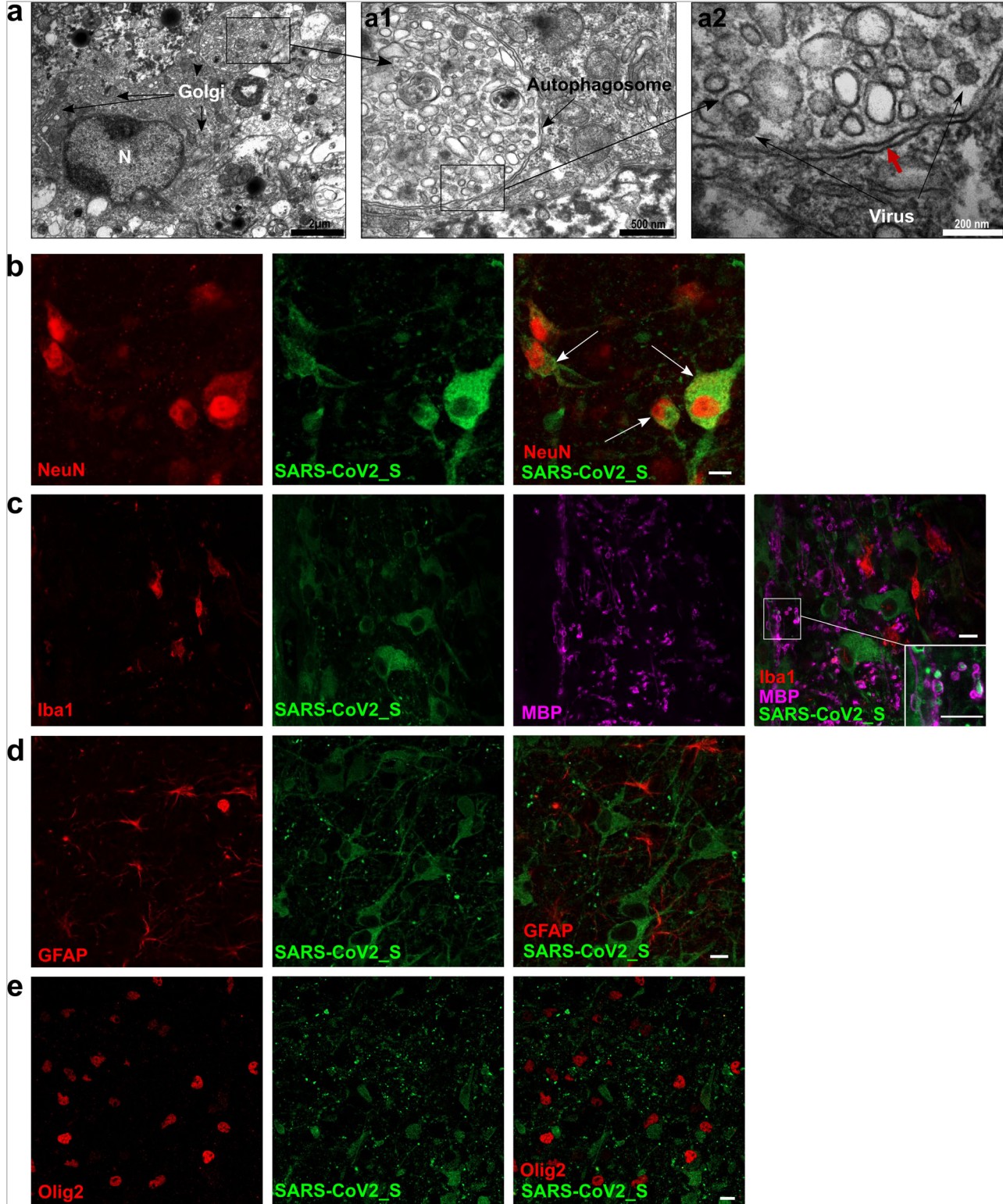

**Fig. 5 SARS-CoV-2 tropism in hamster brainstem organotypic cultures during the first 2 days of infection.** Cultures were infected with 1000 pfu of SARS-CoV-2 and fixed at day 2 days post infection. **a** Transmission electron microscopy (TEM) analysis of a brainstem slice showing an infected neuron with a large Golgi apparatus. **a1, 2** Enlargement of the autophagosome containing viral particles (white arrows). The double membrane of the autophagosome is indicated with the red arrow. The results are representative of two independent experiments. **b–e** Brainstem slices stained with antibodies anti-SARS-CoV-2_S, **b** anti-NeuN, **c** anti-Myelin Basic Protein (MBP), and ionized calcium-binding adaptor molecule 1 (Iba1), **d** anti-glial fibrillary acidic protein (GFAP), and **e** anti-Olig2. The immunofluorescence staining analysis was performed by confocal microscopy and is representative of three independent experiments. Scale bar 10 μm.

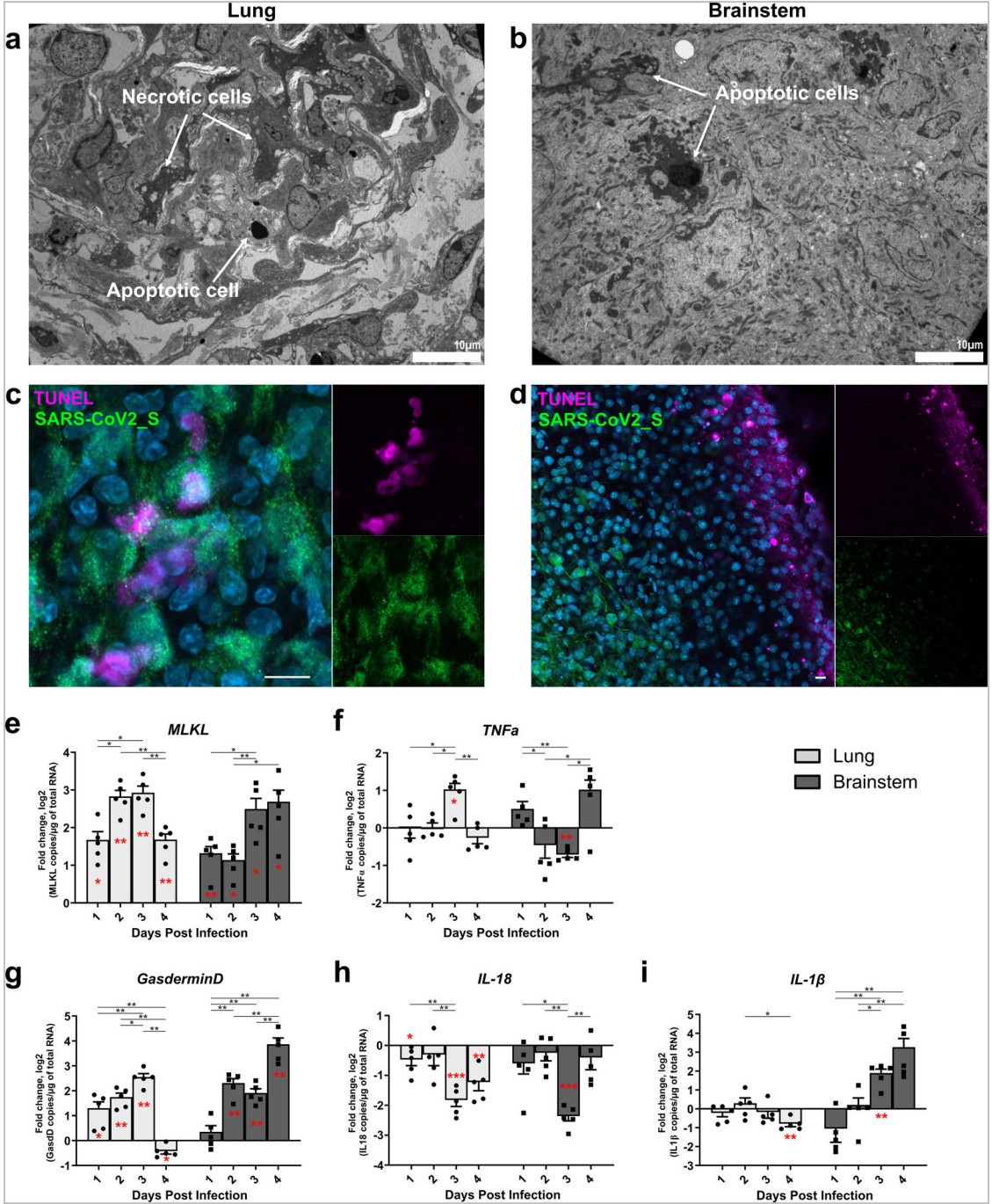

**Fig. 6 Cell death in the ex vivo cultures.** Hamster ex vivo cultures were infected with 1000 pfu of SARS-CoV-2 ($n = 5$ biologically independent animals). **a–c** Lung and brainstem slices were fixed at 1 day post infection (dpi) or 2 dpi, respectively. **a, b** Transmission electron microscopy analysis showing necrotic cells and apoptotic cells, scale bar is represented bottom right on each picture. **c, d** SARS-CoV-2_S protein immunostaining, and terminal deoxynucleotidyl transferase dUTP nick end labeling (TUNEL) labeling in the lung and brainstem. Nuclei were counterstained with DAPI. **e–i** mRNA expression level of **e** *MLKL*, **f** *TNF-α*, **g** *Gasdermin D*, **h** *IL1β*, and **i** *IL-18* over time. mRNA copies per μg of total RNA were quantified by RT-qPCR and normalized to the variation of the amounts of *GAPDH* mRNA. Fold changes are relative to the number of copies of mRNAs in infected organotypic cultures compared to the uninfected ones. Error bars represent SD. Statistical analyses were performed using the Mann–Whitney test two-sided to compare the fold changes between days of culture (black stars). mRNA expression levels in infected samples were also compared with non-infected samples at the corresponding time point (red stars) using the one-sample T-test. *$P < 0.05$; **$P < 0.01$; ***$P < 0.001$. Scale bar 10 μm. Source data are provided as a Source Data file.

organotypic cultures were found to be highly suited for studying viruses able to directly infect the respiratory tract. In line with our previous studies showing that hyperfusogenic neurotropic variants are generally found in the brain[80], the neuro-adapted MeV IC323-EGFP-F L454W did not directly infect lung cultures,

suggesting that additional mechanisms are required to access the Nectin-4 viral entry receptor. We found that ACE2 activity per μg of total protein was similar in the lung and the brain, with the caveats that the assay does not specify which cells express the receptor and to what extent (Fig. 1 and Supplementary Fig. 1) and

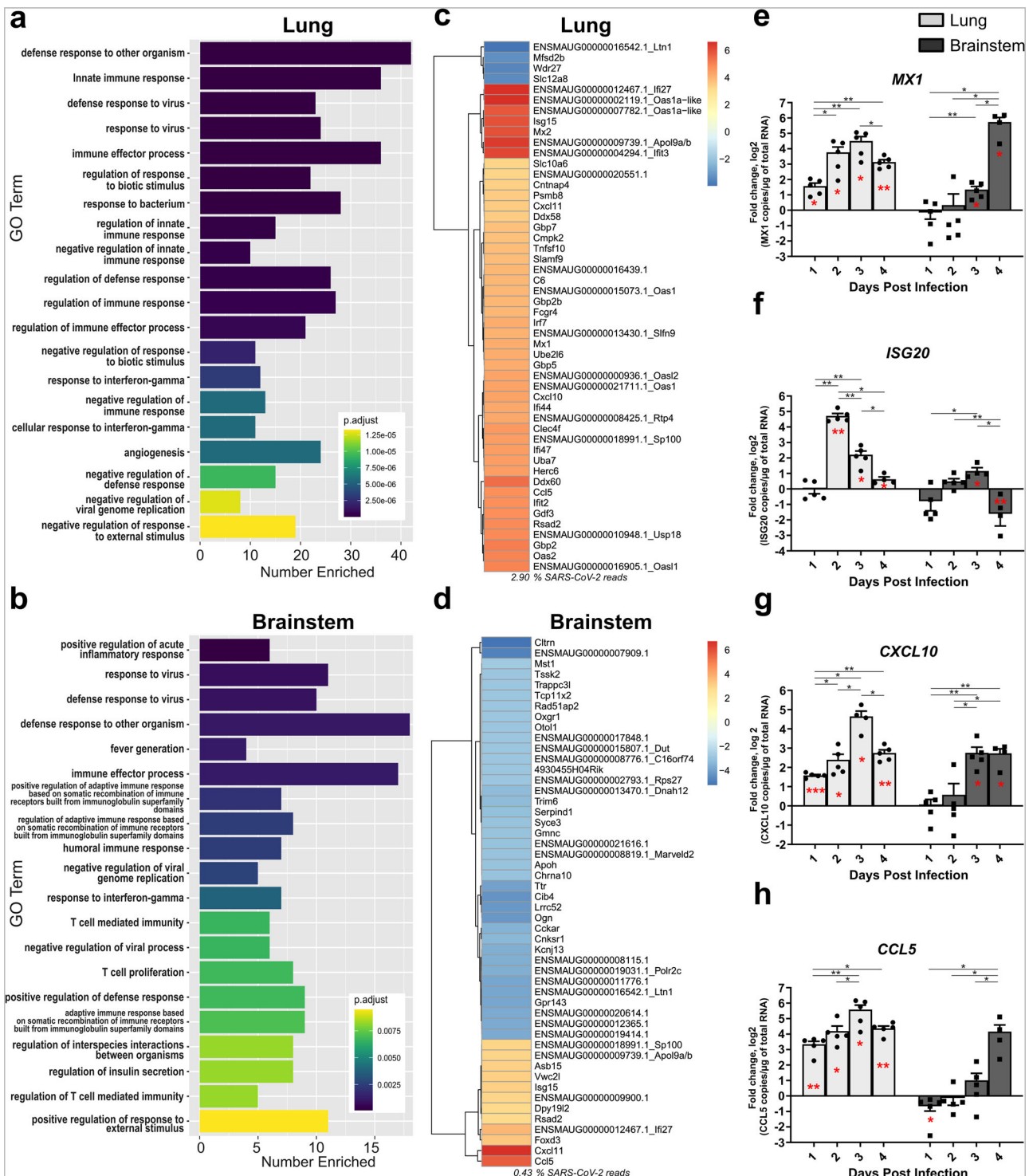

**Fig. 7 Innate immune transcriptional signature in lung and brainstem organotypic cultures during SARS-CoV-2 infection.** Hamster ex vivo cultures were infected with 1000 pfu of SARS-CoV-2 ($n = 5$ biologically independent animals). **a–d** Transcriptomic analysis of the organotypic lung cultures and brainstem cultures 4 days post infection. **a, b** Gene Ontology (GO) analysis. For each tissue, the 20 non-redundant GO categories with the lowest adjusted $p$ value for Fisher exact test of enrichment were displayed using ggplot. **c, d** Heatmaps generated by calculating the log2-fold change for infected samples relative to uninfected samples and taking the 50 with the largest absolute value. **e–h** mRNA expression level of **e** *MX1*, **f** *ISG20*, **g** *CXCL10*, and **h** *CCL5* over time. mRNA copies per μg of total RNA were quantified by RT-qPCR and normalized to the variation of the amounts of *GAPDH* mRNA. Fold changes are relative to the number of copies of mRNA in infected organotypic cultures compared to the uninfected ones. Error bars represent SD. Statistical analyses were performed using the Mann–Whitney test two-sided to compare the fold changes between days of culture (black stars). mRNA expression level in infected samples was also compared with non-infected samples at the corresponding time point (red stars) using the one sample $t$ test. *$P < 0.05$; **$P < 0.01$; ***$P < 0.001$. *MX1* Myxovirus Resistance 1, *ISG20* Interferon Stimulated Exonuclease Gene 20, *CXCL10* C-X-C Motif Chemokine Ligand 10, CCL5: C-C Motif Chemokine Ligand 5. Source data are provided as a Source Data file.

that ACE2 mRNA levels were quite low in both organs. RT-qPCR analysis showed that all receptors and enzymes required for viral entry are expressed in the lungs, suggesting that SARS-CoV-2 could infect the organ either by fusing at the cell surface or through the endosomal pathways. In hamster lungs, a recent study described that both ACE2 and TMPRSS2 expression are required for efficient SARS-CoV-2 infection[81].

In the human lung, ACE2 expression concentrates at the surface of lung alveolar epithelial cells, mainly on type 1 and type 2 pneumocytes but also in bronchial epithelial cells[82]. In the hamster organotypic lung culture model, ciliated cells, type 1 and type 2 pneumocytes are infected (Fig. 4). Our findings are consistent with other in vivo SARS-CoV-2 studies in hamsters that reveal inflammation and severe lesions in the lung with infection of the bronchiolar epithelial cells, type 1 and 2 pneumocytes[45,47]. We also found viral clusters in autophagosomes, confirming that this observation is conserved between infected species (Fig. 4, Fig. 5). In bronchial HAE apically infected with SARS-CoV-2, TEM analysis also highlighted double-membrane vesicles and spherule-containing virions[83].

In distinction to SARS-CoV, it remains unclear whether SARS-CoV-2 targets the brain, and the organotypic models do not allow investigating how the virus spread to CNS. The most commonly observed affection of the CNS seems to be related to immune (/inflammation) response to the infection of choroid plexi or brain vasculature[27]. Multiple alternative routes of entry in the CNS are regularly proposed such as direct entry through the sensorial neurons of the olfactory bulb or cranial nerves (CN)[12,14,46,53]. Secondary infections of the CNS through retrograde transport from sensory neurons (notably the vagus nerve) from initially infected lungs or gut via the enteric nervous system are also commonly suspected[84]. As for other viruses, it remains very difficult to appreciate the real ability of SARS-CoV-2 to infect CNS cells in humans since most of the data are coming post mortem analyses performed mainly on patients who died from acute respiratory syndrome associated with cytokine storm which may strongly affect viral spread to the brain parenchyma. Our data show that, provided that SARS-CoV-2 reaches the brain parenchyma, it can infect CNS cells, initiate transcription (confirmed by neon green expression), and spread in at least two brain substructures (as shown by cell-to-cell dissemination of neon green fluorescent protein) (Fig. 2 and Supplementary Fig. 2). Viral replication is confirmed by the over three logs increase in the number of viral genome copies in a culture in which cell division is very limited (as confirmed by Alarmar blue measurements over time) compared with cell cultures.

Our brainstem cultures derive mainly from the medulla and the pons for the top-central part[85]. SARS-CoV-2 seems to target specific locations in the pons and in the medulla: the motor and sensory areas, respectively. We observed green fluorescence in the zone of X and XII CN fibers, which are responsible for motor (CN X and CN XII) and sensory functions (CN X). In the cerebellar cultures, the infection starts from the deep nucleus and the thin layer of granular cells (Fig. 2 and Supplementary Fig. 2). Interestingly, NiV and MeV infections target quite different areas and propagate more extensively, possibly due to their broader tropism in the CNS. This underlines the interest of organotypic cultures to reveal cell specificity of different viral infections not previously observed in vivo.

At the molecular level, TMPRSS2 expression in the brain tissues was below the quantification limit, which is aligned with the literature that describes a very low expression of TMPRSS2, as well as ACE2, in the brain and neural tissue[53,86,87]. Several studies highlighted the high variation of ACE2 mRNA and protein expression depending on the brain region and established a link with viral tropism[24,88–90]. In the hamster brainstem and

cerebellum, we quantified less than one ACE2 mRNA copy per cell, for all cell types, suggesting that either brain cells expressing ACE2 are very few or that SARS-CoV-2 may not use ACE2 to enter the few CNS target cells when it infects neural tissue.

As expected, Neuropilin-1 transcription in both lung and brain cultures was confirmed[86], with significantly higher mRNA expression levels in the lung than in brain cultures. Cathepsin B and L expressions were similar in both organs. In addition, while cathepsin B and L can replace TMPRSS2 for S cleavage[91], their endosomal distribution may not allow the virus to enter as efficiently as TMPRSS2. We suggest that a lower level of host protease activity in the brain may contribute to the delayed infection by constraining the virus to the alternative endosomal-lysosomal pathway for entry[92]. Another more technical explanation for the faster replication of SARS-CoV-2 in lung cultures is that these organotypic cultures are larger and thicker than brainstem cultures and may contain a broader range and a higher quantity of susceptible cells, accounting for the higher concentration of viral RNA after 2 hours (Fig. 2).

Several studies have shown that SARS-CoV-2 can infect neurons from human induced pluripotent stem cells-derived brain organoids[25,27,29,93]. In hamsters, viral antigens are commonly observed in olfactory neurons, but very few or none in the brain parenchyma, while the virus can be recovered from the brain of a one-month-old infected animal[46,47]. These data suggest that the virus may have limited capacity to cross the blood-brain barrier. Here, we show that neurons from hamster brainstems and cerebella can be infected by SARS-CoV-2 (Fig. 5). As observed by TEM analysis, NeuN staining revealed infection of granular neurons, which are likely to be Golgi II neurons (Fig. 5a,b). Interestingly, we came to the same conclusion with regards to cerebellum cultures in which we did not detect any infection of Purkinje neurons (Supplementary Fig. 4). Although ACE2 expression has been described in neurons and glial cells, we detected fewer than one mRNA copy of ACE2 per cell, and in the conditions tested we did not detect any infection in oligodendrocytes, neither in their cell bodies, nor in myelin fibers. However, myelinated neurons were infected[94]. Neurons are largely nonrenewable and are known to respond moderately to inflammatory and antiviral cytokine stimuli in order to maintain their essential activity[71,95,96]. This may explain at least partially their higher susceptibility to numerous viruses, including SARS-CoV-2. Overall, our results in hamsters but not in wild-type mice suggest an early tropism for specific subtypes of neurons in the CNS, in distinction to the broader range of cells targeted in the lungs.

At the intracellular level, infected lung and brain cultures displayed numerous vacuoles, cytoplasm material degradation, marks of stress, membrane coiling, and disorganized Golgi apparatuses as evidence of cell degeneration or death. Cell death has been evaluated in SARS-CoV-2-infected organs in vivo (i.e., in hamster lungs) and in neurons from human brain organoids[45,93]. Our experiments revealed that apoptosis, necroptosis, and pyroptosis occurred during SARS-CoV-2 infection in both lungs and the brainstem (Fig. 6). Apoptosis was directly observed in both lung and brainstem cells via TEM analysis (Fig. 6), and TUNEL assays showed that very few cells undergoing apoptosis were infected, suggesting that apoptosis may occur through indirect stimuli. Necroptosis, a cell death mechanism involving TNFα stimuli[97,98], appeared in both organotypic cultures, consistent with the uncontrolled expression of TNFα and high levels of MLKL mRNA we found over the 4 days of infection in both the lungs and brainstem, compared to non-infected cultures. A signature of pyroptosis emerged in both ex vivo systems, as indicated by the high levels of Gasdermin D mRNA. However, it has been reported that pyroptosis can be triggered through two distinct pathways, either

through direct pathogen recognition and IL-1β[99] or the activation of ST2 receptors by IL-33[100]. The discordant results we obtained for IL-1β mRNA between the brainstem and the lungs demonstrate tissue-dependent pyroptosis. Although IL-18 mRNA copy number remained low in both organs, the brainstem showed a large increase in IL-1β mRNA levels that are known to be associated with Gasdermin D involvement[101]. In contrast, the levels of IL-1β mRNA remained low despite the upregulation of IL-1β, demonstrating that pyroptosis operates differently in the lungs and the brainstem.

In support of this hypothesis, a previous study demonstrated that high levels of IL-33 were released and ST2 receptors signaling pathways were highly triggered during infection of epithelial airway tissues, provoking further pyroptosis[102]. Altogether, our observations suggest that tissue-associated cell death induced by SARS-CoV-2 infection in both organs could be the result of distinct dysregulated inflammatory responses. Such distinct cytotoxicity may also partially explain the lower representation of SARS-CoV-2 in the human CNS. In this case, the virus may kill neurons it infects at the periphery faster than it spreads. Thus, even a very low level of neuron infection may have important side effects leading to vagus nerve dysfunction or to the initiation of encephalitis[84].

The IFN-I and IFN-III responses were triggered in various in vitro and in vivo models during SARS-CoV-2 infection[103,104]. We confirmed these results in our ex vivo organotypic cultures, showing an increase of both IFN-I and IFN-III-triggered ISGs ISG20 and MX1 mRNA levels mainly involved in establishing optimal antiviral and inflammatory environments when their expressions are correctly balanced (Fig. 7).

In parallel, we demonstrated increased mRNA levels for CXCL10 and CCL5, attractant chemokines that recruit inflammatory mediators. As expected from the delay in the start of viral replication, CXCL10, and CCL5 responses were delayed in the brainstem compared with the lungs, reflecting the discrepancies in cell susceptibility to viral infection in the two tissues.

Apart from allowing better characterizing SARS-CoV-2 infection in the tissue, our organotypic cultures have proved applicable for drug testing. Our data with remdesivir, hydroxychloroquine, and 17-DMAG demonstrated that both hamster lung and brainstem organotypic cultures can serve as efficient tools for screening antiviral drugs and predicting their efficacy and toxicity in vivo. More specifically, the lack of expression of TMPRSS2 in the brainstem suggests that, unlike in the lungs, the S in it is cleaved predominantly by endosomal cathepsins as opposed to cell-surface proteases. As hydroxychloroquine blocks virus-endosome fusion by increasing the endosomal pH[59], our data showing its efficacy in the brainstem but not in the lungs confirm that our model can predict the lack of in vivo effectiveness of a drug. Even in the context of infection with respiratory viruses such as SARS-CoV-2, which do not use the CNS as their main target, brainstem and cerebellum cultures are highly relevant tools for testing drugs that may target the endosomal pathway. The use of both lung and brain cultures to assess the efficacy of antivirals creates an ideal combination for extrapolating their activity in vivo. This could be applied not only against SARS-CoV-2 but also against other respiratory viruses known to be capable of invading the CNS.

In this study, we characterize two ex vivo models to establish their utility for investigating infectious pathogens. We demonstrate that organotypic lung and brainstem cultures are relevant 3D physiological models for several human viral infections, including that by SARS-CoV-2, which are useful for assessing antiviral drugs potentially efficient during the early stages of the infection. These models can also lend themselves to real-time evaluation of emerging mutants. SARS-CoV-2-infected type 1 and type 2 pneumocytes and ciliated cells in the lungs, and granular and Golgi neurons in cerebral structures.

Expression of ACE2, TMPRSS2, and Cathepsin B in organotypic cultures was consistent with the in vivo distribution of SARS-CoV-2 during the first 4 days of infection. We highlight a correlation between TMPRSS2 expression levels and a delay in CNS infection at the organ level, which could offer the innate immune system a window of opportunity for preventing CNS infection. Once SARS-CoV-2 reaches the brain parenchyma, the virus could infect specific neurons potentially involved in respiratory and cardiac function in the brainstem and cerebellum. We describe the induction of type I and III innate immune responses and an inflammatory response to infection at the organ level. Finally, in both organotypic cultures, we observe cell death caused by the infection via caspase-3-independent apoptosis, necroptosis, and pyroptosis.

## Methods

**Viruses**. BetaCoV/France/IDF0571/2020 virus (GISAID Accession ID = EPI_ISL_411218) was isolated in Vero E6 from a nasal swab of one of the first COVID-19-positive patients in France[105] and was kindly provided by the Virpath lab.

2019-nCoV/USA_WA1/2020 virus was isolated by the CDC in the United States, from the first patient diagnosed in the US.

The recombinant NeonGreen SARS-CoV-2 virus (icSARS-CoV-2-mNG) has been generated by introducing the neon green reporter gene into the ORF7 of the viral genome as described elsewhere[54].

The recombinant Measles virus (MeV IC323-EGFP-F L454W) is a hyperfusogenic MeV variant able to disseminate in the absence of known receptors[106]. This variant is expressing the gene encoding EGFP and was generated using reverse genetics in 293-3-46 cells as previously described[107] after modification of the plasmid encoding MeV IC323-EGFP (kindly provided by Yanagi, Kyushu University, Fukyoka, Japan). The recombinant Nipah virus (rNiV-EGFP) is expressing the gene encoding the EGFP and was generated using reverse genetics in 293 cells and prepared as previously described[108]. NiV infections were carried out at the INSERM Jean Mérieux BSL4 laboratory in Lyon, France.

All viruses have been produced and titrated at 37 °C in Vero E6 cells (SARS-CoV-2 and rNiV-EGFP viruses) or in Vero E6-expressing human SLAM receptor (MeV IC323-EGFP-F L454W) (Supplementary Table 1). Briefly, for stock production, cells were infected with MOI = 0.01 in DMEM. After 90 min of incubation at 37 °C, the medium was replaced with DMEM-2% FBS (SARS-CoV-2) or added in order to obtain DMEM-5% FBS (rNiV-EGFP and MeV IC323-EGFP-F L454W), and the cells were incubated at 37 °C in 5% $CO_2$ atmosphere for two days. Viral supernatants were collected and centrifuged ($400 \times g$, 5 min), aliquoted, and titrated as plaque-forming units using a classic dilution limit assay.

**Alamar blue assay**. The organotypic cultures ($n = 6$) were immersed in 200 μl of 1× Alamarblue® (Invitrogen; DAL1025) solution in a culture medium for 2 h at 37 °C in a humidified atmosphere in 5% $CO_2$. The fluorescence was read in a 96-well white plate using 560/590 nm (ex/em) filter settings according to the manufacturer's protocol.

**ACE2 activity assay**. The ACE2 enzymatic activity was quantified using 100 μg of organotypic cultures ($n = 3$) using the ACE2 Activity Assay Kit (Fluorometric) (CliniSciences; K897-100) and following the manufacturer recommendations. Total protein was quantified using the Micro BCA Protein Assay Kit (Thermo Fisher Scientific).

**Animals**. Suckling Syrian golden hamsters (Mesocricetus auratus) used in our study were obtained from Janvier Labs (53940 Le Genest-Saint-Isle, France) with clean health monitoring report. C57BL/6 suckling mice are coming from the animal facility "Plateau de Biologie Expérimentale de la Souris" (PBES) in Lyon.

All animals were used at seven days old. The sex of the animals was random and dependent on the litter threw by the mother.

This study was performed according to French ethical committee (CECCAPP) regulations (accreditation CECCAPP_ENS_2014_034).

**Organotypic culture preparation and treatment**. Organotypic cultures were prepared from suckling hamsters (Mesocricetus auratus, Janvier lab) or C57BL/6 mice (Supplementary Fig. 3) and maintained in culture as detailed elsewhere[50]. In brief, organ substructures (i.e., cerebellum, brainstem, left lung) were isolated from 7-day-old animals ($n = 5$; sex non-discriminated) and cut with a McIlwain tissue chopper (WPI-Europe): 350-μm-thick progressive slices were prepared for brainstems and cerebella, and 500-μm-thick slices for the lungs. The cultures were then separated from each other in cold Hibernate®-A/5 g/L D-Glucose/1× Kynurenic acid buffer and laid out on hydrophilic PTFE cell culture insert membranes (PICM0RG50, Millipore). Slices were subsequently cultured in Minimal Essential

Medium GlutaMAX supplemented with 25% horse serum, 5 g/L glucose, 1% HEPES (all Thermo Fisher Scientific), and 0.1 mg/L human recombinant insulin (Sigma-Aldrich) at 37 °C in 5% $CO_2$. The medium was changed every other day after the slicing procedure. Slices from 5 hamsters were infected on the day of slicing with SARS-CoV-2 (BetaCoV/France/IDF0571/2020), icSARS-CoV-2-mNG[54], MeV IC323-EGFP-F L454W and rNiV-eGFP. For the treatment, cultures were then treated from 90 min post infection to day 4 post infection either with remdesivir (GS-5734; Cliniisciences) diluted in Neurobasal medium or with vehicle (untreated condition) once a day for the 10 µM dose and twice a day for the 2 µM condition. In all, 2 µl of 100 µM or 20 µM of remdesivir were added on top of each of the 5 slices in each well and the remdesivir concentration in the feeding medium was also adjusted in order to reach a final concentration of 10 µM or 2 µM in the insert. The cultures were also treated daily under the same condition with the 17-DMAG (17-Dimethylaminoethylamino-17-demethoxygeldanamycin, Sigma; D5193) at 3 µM and with Hydroxychloroquine sulfate (Sigma; H0915-5MG) at 10 µM and 20 µM. At each time point, slices were collected, and the RNA was extracted in order to perform RT-qPCR.

**RNA extraction and quantitative RT-PCR**. Total RNA from organotypic cultures was extracted using the NucleoSpin RNA Kit (Macherey-Nagel) and quantified with a spectrophotometer (DS-11-FX, DeNovix). For an upcoming SARS-CoV-2 genome quantification, reflecting the viral replication, 100 ng of total RNA were reverse-transcribed using the SuperScript™ III Reverse Transcriptase (Invitrogen) with the SARS-CoV2_tagged primer: 5′-gcagggcaatctca-caatcaggGGTCTGCATGAGTTTAGG-3′ that binds in the intergenic sequence between the N gene and the ORF10. To improve the specificity of the RT-qPCR we have opted for this specific reverse transcription using a primer specific to SARS-CoV-2 genome tagged with a Nipah virus (NiV) derived sequence. The qPCR step uses then a reverse primer against SARS-CoV-2 and a forward primer against NiV extension. To quantify the viral transcription, 100 ng of total RNA were reverse-transcribed using the SuperScript™ III Reverse Transcriptase (Invitrogen) with oligo-dT primers. qPCR was performed using the primers for SARS-CoV-2 Nucleoprotein (Supplementary Table 2).

For the *GAPDH* (Glyceraldehyde 3-phosphate dehydrogenase) and all other genes, 100 ng of total RNA were reverse-transcribed using the iScript cDNA Synthesis Kit (Bio-Rad), based on oligo-dT and random hexamer primers. Obtained cDNAs were diluted 1 : 10. Quantitative PCR was performed using Platinum SYBR Green qPCR SuperMix-UDG (Invitrogen) on a StepOnePlus Real-Time PCR System (Applied Biosystems). Primers were either designed using the Beacon Designer (version 8) software or chosen after validation that their efficacy was close to 100% according to the MIQE checklist[109]. All samples were run in duplicates and results were analyzed using StepOne version 2.3 (Applied Biosystems). All results were normalized to the standard deviation (SD) for *GAPDH* mRNA, and the calculations were performed using the 2∆∆CT model[110]. For each time point, fold changes are relative to the number of copies of mRNAs in infected organotypic cultures compared to the uninfected ones.

**Organotypic culture RNA-Seq and analysis**. RNA from uninfected and infected hamster organotypic cultures (a pool of five slices) was collected and extracted as described above and submitted to the JP Sulzberger Columbia Genome Center for library preparation and sequencing. Strand-specific RNA-Seq libraries were prepared using a poly-A enrichment and were sequenced on an Illumina NovaSeq 6000 with paired-end 2 × 100 reads (Nextera xt kit; Illumina). After quality and adapter trimming with Trimmomatic v0.39[111], transcript abundance quantification was performed using Kallisto version 0.46.0[112] with the Ensembl *Mesocricetus auratus* v1.0 as the reference genome.

Differential gene expression analysis was performed using the Kallisto transcript abundances and the R Bioconductor package DESeq2[113]. In lung cultures, 262 genes out of 15,870 expressed genes were differentially expressed (DE) at a threshold of the absolute value of log2-fold change >2. In brainstem cultures, 170/16041 expressed genes were DE. Code for analysis is available at https://github.com/greninger-lab/SARS-CoV-2_hamster_RNAseq.

Since the hamster genome remains relatively poorly annotated, GO analysis was performed using mouse annotations to test for statistical enrichment of DE genes in GO categories. After the exclusion of genes that did not have a mouse ortholog, 195 DE genes out of 11,726 were analyzed for the lung, and 125/12,698 for the brainstem, using the R package clusterProfiler[114]. For each tissue, the 20 non-redundant GO categories with the lowest adjusted *p* value for Fisher exact test of enrichment were displayed using ggplot, version 3.3.5[115].

To calculate % reads on target for SARS-CoV-2 reads, each sample was aligned against the EPI_ISL_411218 SARS-CoV-2 reference sequence using Bowtie2 with default parameters[116]. SARS-CoV-2 on target percentages were calculated using the number of mapped reads in the resulting BAM file.

Heatmaps were generated using the R package pheatmap by calculating the log2-fold change for infected samples relative to uninfected ones and taking the 50 with the largest absolute value, after eliminating those in which there were fewer than five normalized counts in the uninfected sample for the lungs, or one for the brain. Genes for which the annotation did not include a formal gene symbol were

manually searched in the Ensembl database for highly conserved rodent orthologs, which are included next to the Ensembl gene name in the heatmaps.

**Immunofluorescent staining**. Organotypic cultures from 7-day-old hamsters were infected with 1000 PFU of BetaCoV/France/IDF0571/2020 virus. In all, 24 h post infection, the slices were fixed during 1 h in 4% paraformaldehyde (PFA), washed in 1× Dulbecco's phosphate-buffered saline (DPBS), and permeabilized and blocked in 1× DPBS-3% BSA- Triton X-100 (perm and block solution) overnight at 4 °C. Slices were incubated in the perm and block solution containing the primary antibodies overnight at 4 °C. After three washes (5 min each) in 1× DPBS, slices were incubated in the perm and block solution containing the secondary antibodies for 1 h at RT; donkey anti-rabbit conjugated with Alexa 488 or 555, donkey anti-mouse conjugated with Alexa 488 or 555, and donkey anti-goat conjugated with Alexa 555 or 647 antibodies (1:500 each) (Supplementary Table 3). After 3 washes in 1× DPBS, slices were mounted with Fluoromount-G® aqueous mounting medium (SouthernBiotech, catalog no. 0100-01) on epoxy slides (CEL-LINE, cat-alog no. 30-12A-BLACK-CE24) and coverslipped. Images were taken using an inverted microscope Zeiss Axio Observer. Z1 with confocal unit LSM 800 and analyzed using ImageJ software. All primary antibodies used in this study were validated previously for use in hamster tissue or the sequence homology of the epitopes was more than 85%.

**TUNEL assay**. TUNEL assay (Click-iT™ Plus TUNEL Assay for In Situ Apoptosis Detection, Alexa Fluor™ 647 dye, Thermo Fisher Scientific) was performed following the manufacturer's recommendations.

**Transmission electron microscopy**. Infected organotypic cultures were fixed by immersion in 2.5% glutaraldehyde (Sigma) and 2.5% PFA in cacodylate buffer (0.1 M, pH 7.2) at 4 °C for several days. The samples were post fixed in 1% osmium tetroxide in 0.1 M cacodylate buffer for 1 h at 4 °C, and rinsed with Cacodylate buffer 0.1 M (2 × 10 min) and with water (2 × 10 min) and immersed in uranyl acetate at 4% for 2 h at 4 °C. Samples were dehydrated through graded alcohol (50, 70, 90, and 100%) and propylene oxide for 30 min each and embedded in Epon™ 812 (Sigma-Aldrich, Saint-Louis, Missouri, USA). Semi-thin sections were cut at 2 µm with an ultra-microtome (Leica Ultracut UCT) and stained with 1% Toluidine blue in 1% sodium borate, examined by Leica optical microscope (LEICA DMLB, Leica Microsystems GmbH; Germany). Ultrathin sections were cut at 70 nm and contrasted with uranyl acetate and lead citrate and examined at 70 kv with a Morgagni 268D electron microscope (FEI Electron Optics, Eindhoven, and the Netherlands). Images were captured digitally by Mega View III camera (Soft Imaging System).

**Fluorescence-activated single-cell sorting (FACS)**. Five cerebella from suckling hamster (10 organotypic cultures per cerebellum) were dissociated in a solution of 10 mg/ml of pre-activated papaïn diluted in DMEM containing 10% kynurenic acid. After a 30 min incubation at 37 °C the reaction was stopped by adding FBS. The cells were then split into five separate tubes, washed with Neurobasal medium (Gibco; 12348017), and centrifuged at 400 × *g* for 5 min. Each cell population was stained for 30 min at 4 °C using the following antibodies (Supplementary Table 4). Cells were washed with Neurobasal medium, centrifuged at 400 × *g* for 5 min, and resuspended in 1 ml of Neurobasal medium. The cell was sorted using BD FAC-SAria™ II and collected in 600 µl of RLT with beta-mercaptoethanol. The RNA was then extracted as described above, Reverse transcription was performed using the iScript kit (Biorad) and the mRNA copies of the gene of interest were quantified by RT-qPCR as described above.

**Statistical analysis**. Statistical analyses for figs. 1 and 2 were performed using the Kruskal–Wallis test. ***P < 0.001. For Figs. 4 and 5, statistical analyses were performed using the Mann–Whitney test and the one-sample *t* test. *P < 0.05; **P < 0.01; ***P < 0.001. All statistical analyses were performed in GraphPad Prism5 software.

**Reporting summary**. Further information on research design is available in the Nature Research Reporting Summary linked to this article.

## Data availability
The data generated in this study have been deposited in the Figshare database under accession code https://doi.org/10.6084/m9.figshare.15112443.v1[117]. Source data are provided with this paper.

## Code availability
For the RNA-seq data, code for analysis is available at https://github.com/greninger-lab/SARS-CoV-2_hamster_RNAseq[118].

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

## Acknowledgements

Figure 1a has been created with BioRender.com. We thank Bruno Lina, Andres Pizzorno, and Manuel Rosa-Calatrava from the CIRI, Centre International de Recherche en Infectiologie, LYON, France, for providing us with the BetaCoV/France/IDF0571/2020 virus. We acknowledge World Reference Center for Emerging Viruses and Arboviruses (WRCEVA) and UTMB investigator, Dr. Pei Yong Shi for kindly providing recombinant icSARS-CoV-2-mNG virus based on 2019-nCoV/USA_WA1/2020 isolate. We thank Sonia Longhi from AFMB UMR 7257, CNRS and Aix-Marseille University, France, and Frédéric Carrière from BIP UMR 7281, CNRS and Aix-Marseille University, France, for their advice and help with proofreading of the manuscript. We thank Denis Gerlier for his helpful advice and critical reading of the manuscript. We thank Fabienne Archer for providing us with anti-SP-C and anti-αTubulin antibodies and for her precious advice. We thank Sophie Shyfrin for technical assistance and English language revision, and Géraldine Gourru-Lesimple for assistance in laboratory management. This work was supported by ANR-CoronaPepStop (ANR-20-COVI-000) and Fondation de France to BH and ANRS-COV8-SARSRhinCell to CM.

## Author contributions

Conceptualization, M.F., C.M., D.D. and M.I.; methodology, M.F. and C.M.; formal analysis, M.F., N.A.P.L. and N.M.; investigation, M.F., D.D., V.F., M.M., J.L.W. and R.P.; resources, O.T.; writing—original draft, M.F., C.M. and M.I.; writing—review & editing, A.M., A.L.G., M.P., N.M., O.T. and B.H.; supervision, C.M.

## Competing interests

The authors declare no competing interests.
