## [Peer Review File · Nature Communications]

Reviewer #1 (Remarks to the Author):

Ferren have used recombinant SARS-CoV-2 expressing a fluorescent reporter protein to study infection and replication in organotypic hamster lung and brainstem culture models. The authors show that the models can be used for screening of antiviral compounds, and identify the phenotype and innate immune response of infected cells.

1. The title should reflect that the organotypic cultures were derived from hamsters.
2. Introduction, paragraph 3: the authors should add one sentence referring to animal models, and cite a recent review of the different models that have been described. This would form a bridge to the next paragraph on the hamster model, which has become one of the most important animal models for COVID19.
3. Results - paragraph 1
 - a. title is incorrect (...susceptible to for...).
 - b. I do not agree with the opening statement: SARS-CoV-2 mainly targets the respiratory tract (but in many individuals the infection remains localized to the URT, resulting in subclinical infection).
 - c. The authors claim that the virus can infect the brainstem, but do not explain how the virus is thought to get access to that tissue from the respiratory tract.
 - d. The authors do not explain the rationale for growing brain stem tissue at air-liquid interface.
 - e. Quantification of the entry receptor addresses susceptibility, not permissiveness
 - f. Final sentence: ...might be less susceptible to...
 - g. Spelling error in fig 1d (Catheptin B)
4. Results – paragraph 2
 - a. Title: ...susceptible and permissive...
 - b. Line 2: add reference in which this recombinant virus was first described.
5. Results – paragraph 3

In order to interpret these data, dose-response curves showing activity of Remdesivir against SARS-CoV-2 in immortalized cells and organotypic brain slices should be shown in parallel. Another useful experiment would be to screen the effect of hydroxychloroquine in cells and the organotypic culture models in parallel, to investigate if this model could have predicted the lack of in vivo effectiveness of HCQ. This figure as now included seems to be a pilot experiment, but does not allow interpretation of the value of this model for screening antiviral compounds.
6. Results -paragraph 4

The authors describe ... multiple signs of cell degradation... They should report if they observed a difference between infected and non-infected cells. In other words: is this a property of the organotypic model system, or of the infection?
7. Results – paragraph 5 and 6

Similar to above: the authors should link their description of cell death signature to virus infection or organotypic cell culture conditions. Were differences between infected and uninfected cultures systematically compared? The results described in paragraph 6 are a great addition to the manuscript.

Reviewer #2 (Remarks to the Author):

The major claims of the paper are that an organotypic culture system has been developed to investigate SARS-CoV 2 infection of the the lung, brainstem and cerebellum. The findings in lung

are as expected, however the findings in brain slices are interesting, and provocative. There is an enormous difference between lung and brain by fluorescence. The quality of the fluorescence in brain is not especially convincing and would need multiple controls for autofluorescence. Furthermore, the discrimination between a labelled virus attached to cells, or taken up by vesicles is different from cells actually replicating virus. The list of primers shows primers to detect both genomic and subgenomic RNAs. The subgenomic PCR would be much more accurate in detecting bona fide virus replication, particularly in brain. It is unclear in which studies the subgenomic PCR is used. If virus peaks at Day 2 in the brain cultures, why is day 4 chosen for the remdesivir experiments. The concentration of remdesivir at 2 and 10 micro molar seems to show a reduced effectiveness relative to previous studies where the EC90m was 1.76 uM. <https://www.nature.com/articles/s41422-020-0282-0>. It would be important to show the expression of virus in brains of infected golden Syrian hamsters, and confirm the relevance of the findings in the organotypic cultures.

Changes in gene expression in the brain may be due to exposure of the tissue specimens to virus preparations rather than replication per se. The extent of virus in vacuoles relative to cell replication in brain would support this notion. It would be important to determine and report the relationship between the input PFU and viral genomes detected by RT PCR in order to evaluate a true replicative capacity.

Reviewer #3 (Remarks to the Author):

This study by Ferren et al. aims to develop two organotypic models of SARS-CoV-2 infection of lung and brainstem, utilizing tissues acquired from naïve 7-day old Syrian golden hamsters. Cerebellum is also investigated, which is somewhat confusing as to the specification of the brainstem as the only CNS structure of relevance in the title and manuscript.

The manuscript is well-written and organotypic models can be very useful for understanding and tweezing out relevant pathways involved in pathogenesis. The organ substructures investigated demonstrate ACE2 activity and transcription of proteases known to cleave the viral spike protein, which would theoretically allow for cell entry of the virus. Using a high SARS-CoV-2 titer, the authors do demonstrate that lung, brainstem, and cerebellum organotypic cultures are susceptible to ex vivo infection. While the lung infection is demonstrated to a similar, albeit lesser, degree in vivo, it's less clear if the considerable infection of the CNS slices has biological relevance, as this does not appear to be seen in human subjects and reduces the enthusiasm for this work. The referenced manuscript of virus infection in brainstem shows an extra-axonal structure suggestive of a single virion. An additional ultrastructure image of the gyrus rectus shows three virus-like particles in a neuron that is damaged, which raises the important question as to how the virus actually entered this cell and whether this is a replication competent environment for the virus. This is an important consideration, as exposing organotypic brain slices to high viral titers, which are likely much greater than that which the brain would be subject to in vivo, can obfuscate neuropathogenesis.

A control that may help to address this major concern may be to expose the CNS slices to high titers of a non-neurotropic virus (similar to what was done with the MeV IC323-EGFP-F L454W assay).

IHC and/or ISH of ACE2, TMPRSS2, and Cathepsin B would help in identification of cells specifically expressing these important antigens. Whole tissue protein and RNA analyses do not provide sufficient support for the capacity of the virus to infect specific cell populations. This is especially relevant with regard to the brain, where ACE2 expression by neurons remains controversial and may vary by brain region.

Immunofluorescent images are very difficult to interpret and without non-infected controls. The colocalization images shown with and without merge may be helpful for interpretation. Viral structure is difficult to determine in TEM images, which makes these panels difficult to assess. TEM panels need visible scale bar to aid interpretation. Some TEM panels are difficult to determine the structure being pointed to. Unbiased stereological quantitation of infected cells may also help interpretation. This is particularly relevant to the CNS structures.

In the paragraph starting at line 244, the mixing of lung and brain discussion does not flow properly and was initially confusing. The statement made at line 260 is not accurate.

No details are provided on the control tissues regarding how these were defined experimentally. How many controls were used? Were they maintained under similar conditions? Were they infected using the media from non-infected Vero E6 cells cultured for the same length of time as cells used for developing the viral stocks? Were they mock treated using vehicle for remdesivir?

Methods should disclose the antibodies with clone and source for immunofluorescent staining. Likewise, additional protocol details of slice infections would be helpful (e.g., length of exposure). Titers of each virus used should be disclosed in the Methods section.

Is the SARS-CoV-2 infection pfu of 10,000 accurate in ED Fig. 1?

RESPONSE TO REFEREES

We thank the referees for their very encouraging comments that allowed improving the understanding of both our models and SARS-CoV-2 early infection at the organ level.

Reviewer #1 (Remarks to the Author):

Ferren have used recombinant SARS-CoV-2 expressing a fluorescent reporter protein to study infection and replication in organotypic hamster lung and brainstem culture models. The authors show that the models can be used for screening of antiviral compounds, and identify the phenotype and innate immune response of infected cells.

1. The title should reflect that the organotypic cultures were derived from hamsters.

The title has been modified accordingly.

2. Introduction, paragraph 3: the authors should add one sentence referring to animal models, and cite a recent review of the different models that have been described. This would form a bridge to the next paragraph on the hamster model, which has become one of the most important animal models for COVID19.

We have added a sentence in the paragraph 3 of the introduction and cited a review from March 2021 (ref n°29; doi:10.1016/j.coviro.2021.03.009) to assess this relevant point.

Line 73 : "To date, different animal models have been described to study SARS-CoV-2 pathogenesis, transmission or antiviral efficacy, including transgenic mice expressing human ACE2, hamsters, ferrets, rhesus macaques, cynomolgus macaques and African green monkeys²⁹."

3. Results - paragraph 1

a. title is incorrect (...susceptible to for...).

b. I do not agree with the opening statement: SARS-CoV-2 mainly targets the respiratory tract (but in many individuals the infection remains localized to the URT, resulting in subclinical infection).

The text has been modified accordingly to the remark 3a and updated as suggested in 3b in order to be more accurate.

Line 97 : "Hamster lung and brainstem *ex vivo* cultures are viable and susceptible to SARS-CoV-2 infection."

Line 99 : "Since SARS-CoV-2 notably targets the upper respiratory tract and lungs and may also infect the brainstem, we have developed new *ex vivo* models of these organs from naive suckling hamsters, based on our previous experience using organotypic cerebellum cultures."

c. The authors claim that the virus can infect the brainstem, but do not explain how the virus is thought to get access to that tissue from the respiratory tract.

How SARS-CoV-2 can reach the central nervous system (CNS) during natural infection remains unclear. Several groups suggested that SARS-CoV-2 may reach the brain through different pathways: through nerves, blood circulation and endothelial cells or simply by crossing the olfactory epithelium to reach the olfactory bulb. We agree that this information improves the

June 2nd 2021

manuscript so we have added several sentences to clarify this point (end of paragraph 1 in the introduction). Irrespective of the route of infection we propose that our study established the hamster brainstem infection with SARS-CoV-2 as a valid model to address the SARS-CoV-2 CNS pathogenesis.

Lines 52 to 58 : “However, the susceptibility of human neurons to the infection and the permissiveness of human brain organoids have been demonstrated *in vitro*^{14–18} and SARS-CoV-2 viral particles or RNA have already been found in the cerebrospinal fluid¹⁹ and in the brain of patients^{11,20}. In order to reach the CNS, SARS-CoV-2 may travel from the periphery into the CNS through the olfactory neurons or through the vagus nerve. In addition, SARS-CoV-2 infection has been shown to disrupt the blood-endothelial barrier by damaging the choroid plexus epithelium and as a consequence of cytokine storm and systemic inflammation^{18,21,22}.

d. The authors do not explain the rationale for growing brain stem tissue at air-liquid interface.

We apologies for the confusion, we did not grow stem tissue but prepared fresh slices of brain directly from the animal and maintained it in culture for several days as formerly developed by neurobiologists for other structures such as cerebellum. As slices are prepared directly from the hamster organs the respiration is privileged compared to cell cultures, as previously described by other authors (ref n°43, doi: 10.1016/j.neuroscience.2015.07.086). and slices need to be fed from the liquid available in the lower compartment, explaining the air-liquid interface necessity. We added a sentence to explain this rationale in the first paragraph of the results section.

Line 103 : “The 3-dimensional cultures were then maintained on a polytetrafluoroethylene (PTFE) membrane in order to keep an air-liquid interface for up to 4 days (Fig. 1a). In opposition with primary cultures, organotypic cultures are usually not fully soaked in medium in order to allow oxygenation. The 0.4 µm semipermeable pores of the insert permit the diffusion of the medium into the cultures.”

e. Quantification of the entry receptor addresses susceptibility, not permissiveness

f. Final sentence: ...might be less susceptible to...

g. Spelling error in fig 1d (Catheptin B)

We thank the reviewer for pointing these mistakes, which we have corrected accordingly.

4. Results – paragraph 2

a. Title: ...susceptible and permissive...

b. Line 2: add reference in which this recombinant virus was first described.

We have modified and we added the needed reference (ref n°45; doi: 10.1016/j.chom.2020.04.004).

5. Results – paragraph 3

In order to interpret these data, dose-response curves showing activity of Remdesivir against SARS-CoV-2 in immortalized cells and organotypic brain slices should be shown in parallel. Another useful experiment would be to screen the effect of hydroxychloroquine in cells and the organotypic culture models in parallel, to investigate if this model could have predicted the lack of in vivo effectiveness of HCQ. This figure as now included seems to be a pilot experiment, but does not allow interpretation of the value of this model for screening antiviral compounds.

We are grateful to the reviewer for these suggestions which allowed us to improve the manuscript. As we have been limited in the number of animals available for the research, due to the recent world penury of hamsters, largely used for COVID-19 we could not perform a dose-response curve in the organotypic cultures with Remdesivir, which required the increased

number of animals. Nevertheless, we had previously determined quite closely effective and ineffective doses in hamster organotypic cultures. Similar analyses have already been reported using several immortalized cell lines in shorter experiments lasting maximum 48 hours (ref. n° 47, doi : 10.1016/j.antiviral.2020.104878; ref n°52, doi : 10.1038/s41422-020-0282-0). In contrast, our cultures are organ slices, composed of a multilayer of cells (approximately 10 or more depending on the thickness) were kept in culture much longer and the viral replication was determined after 4 days. For these reasons we believe comparison between the dose-responses in immortalized cells and our complex 3D organotypic cultures cannot be directly compared. As the reviewer noted molecules showing antiviral efficacy *in vitro* do not work efficiently *in vivo*. As suggested the organotypic may better reflect and model the *in vivo* situation, rather than only reproduce what has been previously shown in *in vitro* cultures. As requested we treated the organotypic cultures with 10µM and 20µM of hydroxychloroquine and our data demonstrated the lack of efficacy of the HCQ to block the infection in the lung cultures (Fig 3d and e). Considering the fact that HCQ blocks the viral entry at the endosomal level, these data are consistent with the literature (ref. n°47, doi : 10.1016/j.antiviral.2020.104878; ref n° 50, doi : 10.1016/j.intimp.2020.107232; ref n°52, doi : 10.1038/s41422-020-0282-0; ref n°53, doi : 10.1093/cid/ciaa237). In order to demonstrate the possible application of the organotypic cultures in the evaluation of possible anti-covid-19 drugs we also tested the effect of 17-DMAG, an HSP90 inhibitor known to be efficient *in vitro* but to exhibit toxicity *in vivo*. Based on RNA quantifications, the results showed that the treatment with 17-DMAG was already toxic on our cultures before showing antiviral efficacy. All together these data provide strong support to the notion that the hamster organotypic lung and brainstem cultures are likely predictive of a drug “true” antiviral activity as well as its toxicity before testing it *in vivo*.

6. Results -paragraph 4

The authors describe ... multiple signs of cell degradation... They should report if they observed a difference between infected and non-infected cells. In other words: is this a property of the organotypic model system, or of the infection?

The multiple signs of cell degradation were indeed compared to the non-infected cultures but we did not include them initially in the TEM images. Therefore, we have added a supplementary Figure presenting non-infected cultures (Extended Data Fig. 5) in the revised manuscript to confirm that the differences occurred in response to the infection. In the non-infected cultures we did not notice any sign of cell degradation at the indicated time.

7. Results – paragraph 5 and 6

Similar to above: the authors should link their description of cell death signatures to virus infection or organotypic cell culture conditions. Were differences between infected and uninfected cultures systematically compared? The results described in paragraph 6 are a great addition to the manuscript.

We apologize if it was not clear. We confirm that for each day all the RT-qPCR data showed in Fig. 6 and 7 are systematically presented in fold change compare to the corresponding non-infected cultures. We have added the sentences “compared to the non-infected one”; “compared to the uninfected ones” and “compared to the respective uninfected conditions”, in the results section, in the methods section and in the discussion section to clarify this point (lines 283, 298, 594, 613, 697 and 724).

June 2nd 2021

Reviewer #2 (Remarks to the Author):

The major claims of the paper are that an organotypic culture system has been developed to investigate SARS-CoV 2 infection of the the lung, brainstem and cerebellum. The findings in lung are as expected, however the findings in brain slices are interesting, and provocative. There is an enormous difference between lung and brain by fluorescence. The quality of the fluorescence in brain is not especially convincing and would need multiple controls for autofluorescence.

We are grateful to the reviewer for this comment and the constructive suggestions. We apologize for the poor quality of the figures in the initial submission, due to the PDF conversion. We have made sure that the high-quality pictures will be associated to the new submission. As an alternative here is a secure link to access to the high quality figures :

<https://filesender.renater.fr/?s=download&token=433287e7-df62-4327-b149-783414a72929>

We agree data are more convincing when additional controls are presented. We modified the Figure 2 so the readers can see the viral dissemination from day 1 up to day 4, side by side and we added a supplementary figure (Extended Data Fig. 3) showing numerous controls clearly excluding the autofluorescence.

Furthermore, the discrimination between a labelled virus attached to cells, or taken up by vesicles is different from cells actually replicating virus. The list of primers shows primers to detect both genomic and subgenomic RNAs. The subgenomic PCR would be much more accurate in detecting bona fide virus replication, particularly in brain. It is unclear in which studies the subgenomic PCR is used.

We apologize for the misunderstanding. To clarify this point we have added in the revised manuscript the reference of the initial publication explaining how the virus was made (ref n°45; doi: 10.1016/j.chom.2020.04.004). The virus we generated is encoding for the mNeon green however it is not fluorescent by itself since extremely little quantity of fluorochrome can be accidentally co-encapsulated in the viral particle. Therefore, in order to exclude any confusion we added images of hamster cultures 2 hours after infection with 10,000 PFU of SARS-CoV-2 (Extended Data Fig 3a) showing no fluorescence. In order to get NeonGreen expression in infected cells, the viral genome needs to be released in the cytoplasm, with the initiation of the transcription and translation.

In order to be more convincing regarding the viral replication we added additional information in the method section on the primer we used for the reverse transcription prior to quantification of the genomic RNA (in the intergenic sequence between the N gene and the ORF10) (Line 681) We also added more data and quantified the viral transcription by doing a reverse transcription using Oligo-dt and then we quantified SARS-CoV-2 N transcript (Fig. 2e,j). A schematic presentation was added (Extended Data Fig. 6) to clearly present all the steps we performed in order to amplify selectively the viral genome and/or SARS-CoV-2 N mRNA. To exclude the confusion we explicitly wrote “replication” or “transcription” on the top of the graphs for the Figure 2. Regarding the antiviral efficacy of Remdesivir, we quantified the genomic N RNA copies from the new cDNA obtained from the specific reverse transcription but from the same RNA and the inhibition of the infection was similar in both quantifications. To ensure consistency, we used these latest data in the Fig. 3a.

If virus peaks at Day 2 in the brain cultures, why is day 4 chosen for the remdesivir experiments.

We thank the reviewer for highlighting this interesting point. The objective was to show that when other can block the infection after 2 days in classic monolayer cultures, here we were even able to block longer in complex organotypic cultures when the treatment started early

June 2nd 2021

enough after infection. The longer period is advantageous as it excludes any residual viral activity which could be amplified in later time points. We have added a sentence in the main text of the results section to assess this question.

Line 186 : “The treatment started two hours after infection (Fig. 3a) and continued for up to 4 dpi to make sure that the treatment not only delays the infection but also blocks viral dissemination.”

The concentration of remdesivir at 2 and 10 micro molar seems to show a reduced effectiveness relative to previous studies where the EC90m was 1.76µM.
<https://www.nature.com/articles/s41422-020-0282-0>.

This is an interesting comment which is also partially explain low *in vivo* efficacy of drug candidate highly potent *in vitro*. In the literature the efficacy has been shown in a monolayer of cells and only 48 hours after the infection. Considering the complexity of our model (3D with an average of 10 layers of cells at least), the air liquid interface similarly to real lung *in vivo* and the fact that we stopped the experiment at 4 days post infection our results clearly highlight the huge gap between *in vitro* efficacy and *in vivo* soundness of a treatment. We have added additional references and we have improved the text to better explain this point (section 3 of the results).

Line 183 : “The slices were treated daily at two different concentrations of remdesivir, one right over the IC90 in cell culture and one five times higher, as our organotypic cultures are more complex than regular monolayer cultures.”

It would be important to show the expression of virus in brains of infected golden Syrian hamsters, and confirm the relevance of the findings in the organotypic cultures.

We have added more references chosen among the numerous studies recently published and showing the virus in hamster’s brain (e.g. ref n° 37, doi:10.1016/j.bbi.2020.06.032 and 38, doi.org/10.1038/s41586-020-2342-5) as well as in human brain. These references confirm brain infection during *in vivo* infection and support the importance of the presented organotypic model.

e.g. : Line 78, “The pathogenesis of SARS-CoV^{33,34} and SARS-CoV-2³⁵⁻³⁸ in Syrian hamsters is similar to that observed in humans, supporting the use of hamsters as models for studying these infection^{36,39,40}”.

Changes in gene expression in the brain may be due to exposure of the tissue specimens to virus preparations rather than replication per se. The extent of virus in vacuoles relative to cell replication in brain would support this notion.

For each experimental point the RT-qPCR data presented in the Fig. 6 and 7 are showed in fold changes compared to the corresponding non-infected cultures. We have added the sentences “compared to the non-infected ones”; “compared to the uninfected ones” and “compared to the respective uninfected conditions”, in the results section, in the methods section and in the discussion section to clarify this point (lines 250 283, 298, 594, 613, 697 and 724). This is also the case for the transcriptomic analysis that present only the genes that are differently expressed from the non-infected cultures after 4 days of culture compared to the 4 days post infection.

June 2nd 2021

It would be important to determine and report the relationship between the input PFU and viral genomes detected by RT PCR in order to evaluate a true replicative capacity.

We quantified the viral genomes per μg of total RNA 2 hours post infection (day zero). These values represent the input PFU that was initially attached to the organotypic cultures. Any increase of the genome numbers during following days of infection demonstrates the productive SARS-CoV-2 replication.

Reviewer #3 (Remarks to the Author):

This study by Ferren et al. aims to develop two organotypic models of SARS-CoV-2 infection of lung and brainstem, utilizing tissues acquired from naïve 7-day old Syrian golden hamsters. Cerebellum is also investigated, which is somewhat confusing as to the specification of the brainstem as the only CNS structure of relevance in the title and manuscript.

We thank the reviewer for the constructive suggestions Cerebellum organotypic cultures have been formerly characterized in our laboratory for other virus infections and in the current manuscript they offer a reference for comparison with the previously published data. Additionally, this substructure of the brain contains a very broad spectrum of neurons and other neural cells thus, providing relevant controls to support and to confirm the susceptibility of neurons which may not exist in other brain area. We mainly focused on the brainstem since numerous studies were suggesting that cranial nerve X as well as brainstem may be targeted directly or indirectly in the pathogenesis of SARS-CoV-2. Here, the main goal was to reply to the question of the structure and cell type early susceptibility in case the virus would reach the area.

The manuscript is well-written and organotypic models can be very useful for understanding and tweezing out relevant pathways involved in pathogenesis. The organ substructures investigated demonstrate ACE2 activity and transcription of proteases known to cleave the viral spike protein, which would theoretically allow for cell entry of the virus.

Using a high SARS-CoV-2 titer, the authors do demonstrate that lung, brainstem, and cerebellum organotypic cultures are susceptible to ex vivo infection.

While the lung infection is demonstrated to a similar, albeit lesser, degree in vivo, it's less clear if the considerable infection of the CNS slices has biological relevance, as this does not appear to be seen in human subjects and reduces the enthusiasm for this work.

We thank the reviewer for these comments. Our main idea was to demonstrate what would occur if the virus was reaching the brain parenchyma. While we already cited few references demonstrating the occurrence of SARS-CoV-2 CNS infection in human, very few were published at the moment of our initial submission. We have added additional recent references in the first paragraph of the introduction notably in high impact journals demonstrating clearly it can occur in human (ref. n°11, doi: 10.1038/s41593-020-00758-5; n°14, doi: 10.1007/s10072-020-04575-3; n°15, doi: 10.3390/v12091004; n°16, doi: 10.1038/s41422-020-0390-x; n°17, doi: 10.14573/altex.2006111; n°18, doi: 10.1016/j.stem.2020.10.001, n°19, doi: 10.1016/j.ijid.2020.03.062; n°20, doi: 10.1101/2020.06.25.169946). (Lines 47 to 54)

The referenced manuscript of virus infection in brainstem shows an extra-axonal structure suggestive of a single virion. An additional ultrastructure image of the gyrus rectus shows three virus-like particles in a neuron that is damaged, which raises the important question as to how the virus actually entered this cell and whether this is a replication competent environment for the virus. This is an important consideration, as exposing organotypic brain slices to high viral titers, which are likely much greater than that which the brain would be subject to in vivo, can obfuscate neuropathogenesis.

We are grateful to the reviewer for this comment. We have added a sentence in the first paragraph of the introduction to clarify how SARS-CoV-2 may reach the brain through different pathways, *in vivo* and *in vitro*.

June 2nd 2021

Lines 49 to 57 : “It has been suggested that SARS-CoV-2 reaches the medulla oblongata and that brainstem infection may be involved in both respiratory and heart failure in patients^{7–11}. To date, the neuro-invasive potential of SARS-CoV-2 in humans remains poorly understood^{12,13}. However, the susceptibility of human neurons to the infection and the permissiveness of human brain organoids have been demonstrated *in vitro*^{14–18} and SARS-CoV-2 viral particles or RNA have already been found in the cerebrospinal fluid¹⁹ and in the brain of patients^{11,20}. In order to reach the CNS, SARS-CoV-2 may travel from the periphery into the CNS through the olfactory neurons or through the vagus nerve. In addition, SARS-CoV-2 infection has been shown to disrupt the blood-endothelial barrier by damaging the choroid plexus epithelium and as a consequence of cytokine storm and systemic inflammation^{18,21,22}.”

Point concerning the possible viral replication in a neural environment was answer in this study showing that the brain cultures from brainstem and cerebellum are both susceptible and permissive to SARS-CoV-2 infection.

Considering the fact that there are at least 10^7 cells within the organotypic slice, the multiplicity of infection (MOI) we used for the infection remains rather low (MOI<0.001 for staining and MOI<0.0001 for most of the other experiments). We have added a control in a supplementary figure (Extended Data Fig. 3) showing icSARS-CoV-2-mNG replication in lung and brain cultures when infection is performed with only 100 PFU, presenting a MOI lower than 0.00001.

A control that may help to address this major concern may be to expose the CNS slices to high titers of a non-neurotropic virus (similar to what was done with the MeV IC323-EGFP-F L454W assay).

We thank the reviewer for this suggestion, however we do not have in the lab a non-neurotropic virus infecting hamster. Nevertheless, in order to answer this point in the other direction we have added an additional control in a supplementary figure (Extended Data Fig. 3). We infected lung, brainstem and cerebellum cultures from C57BL/6 mice (non-susceptible to SARS-CoV-2 infection) with 10,000 PFU of icSARS-CoV-2-mNG and confirmed a complete absence of infection at 1 day post infection (dpi) and at 4 dpi (Extended Data Fig. 3).

IHC and/or ISH of ACE2, TMPRSS2, and Cathepsin B would help in identification of cells specifically expressing these important antigens. Whole tissue protein and RNA analyses do not provide sufficient support for the capacity of the virus to infect specific cell populations.

We added a supplementary figure (Extended Data Fig. 1) with the quantification of mRNA expression of ACE2, TMPRSS2 and Cathepsin B and we also added Neuropilin-1 and Cathepsin L in the study. Unfortunately, as hamster-specific antibodies are not available on the market, after multiple trials we did not find any cross-reactive antibodies good enough to work in hamster for staining. We hope the growing work performed on hamster will help in future to solve these technical issues related to the still too poor number of tools available in hamster. In order to propose an alternative way to answer this question we have sorted the 4 cell populations of the brain of suckling hamsters and quantified the mRNA expression of ACE2, Neuropilin-1, TMPRSS2 and Cathepsin B and L in each cell type (Extended Data Fig. 1).

This is especially relevant with regard to the brain, where ACE2 expression by neurons remains controversial and may vary by brain region.

We fully agree with this comment and our results suggest that the expression of ACE2 remains extremely low in neurons or that very few neurons express ACE2 in our slices. We discuss this

June 2nd 2021

point in the new manuscript (discussion section) and we have added more references (ref n°15, doi: 10.3390/v12091004; n°78, doi : 10.1101/2020.04.19.049254; n°79, doi: 10.1056/NEJMc2011400; n°80, doi: 10.1038/s41421-021-00249-2).

Line 373 : “Several studies highlighted the high variation of ACE2 mRNA and protein expression depending on the brain region and established a link with viral tropism^{15,78–80}.”

Immunofluorescent images are very difficult to interpret and without non-infected controls. The colocalization images shown with and without merge may be helpful for interpretation. Viral structure is difficult to determine in TEM images, which makes these panels difficult to assess. TEM panels need visible scale bar to aid interpretation. Some TEM panels are difficult to determine the structure being pointed to. Unbiased stereological quantitation of infected cells may also help interpretation. This is particularly relevant to the CNS structures.

We apologize for the poor quality of the figures in our initial manuscript, due to the PDF conversion. We provide the revised manuscript with high-quality images, that we expect to be more convincing. As an alternative here is a secure link to access to the high quality figures : <https://filesender.renater.fr/?s=download&token=433287e7-df62-4327-b149-783414a72929>

The immunofluorescence images were split by channel and the figures were edited to show the pictures with and without the merge (Fig. 4 and 5).

Regarding the TEM images, we added larger scale bar to make it more visible (Fig. 4, 5 and 6). We also added a supplementary Figure (Extended Data Fig. 5) with some TEM images from non-infected cultures to help for the interpretation.

In the paragraph starting at line 244, the mixing of lung and brain discussion does not flow properly and was initially confusing. The statement made at line 260 is not accurate.

We agree with this suggestion and have reorganized the text in order to split the lung and the brain in this part of the discussion (Fig. 4 and 5).

No details are provided on the control tissues regarding how these were defined experimentally. How many controls were used? Were they maintained under similar conditions?

We have added this information in the manuscript and in the method section (paragraph “Organotypic culture preparation and treatment”). Briefly, we used minimum 5 controls per conditions and these non-infected cultures were treated under the same conditions as the infected ones (infected with a vehicle).

Were they infected using the media from non-infected Vero E6 cells cultured for the same length of time as cells used for developing the viral stocks?

In this study, slices have received the cell culture medium as a vehicle. We have recently tested the effect of clarified, filtered two days old Vero-E6 supernatant fluid on slice responses and did not detect any effect compared to classic DMEM 2% FCS in terms of infection and chemokine response. We would like to underline that the infections were performed using only 2 to 10 µl of medium. That can explain this absence of effect/difference since the slices were exposed to very small volumes of culture media, which is certainly not enough to trigger visible responses.

June 2nd 2021

Were they mock treated using vehicle for remdesivir?

Yes, we confirm the infection controls are made with supernatant from the same VERO-E6 cell cultures and the non-treated cultures were mock-treated with the vehicle containing the same quantity of DMSO used to solubilize the remdesivir. We specify this in the manuscript (results section and method section).

Line 187 : “Mock organotypic cultures were maintained under similar conditions and treated using a vehicle.”

Line 666 : “For the treatment, cultures were then treated from 90 min post infection to day 4 post infection either with remdesivir (GS-5734; Clinisciences) diluted in Neurobasal medium or with vehicle (untreated condition) once a day for the 10 μ M dose and twice a day for the 2 μ M condition.”

Methods should disclose the antibodies with clone and source for immunofluorescent staining.

We thank the reviewer for noticing that point. We have provided initially the list in an additional document checklist for Nature communications. We have added this information in the method section of the revised version of the manuscript.

Line 743 : “Table 3 : Antibodies used for Immunofluorescent staining”.

Likewise, additional protocol details of slice infections would be helpful (e.g., length of exposure).

Infections are performed by adding drops of 2 μ L of virus stock at the right concentration on the top of each slice. These drops are absorbed by the slices in few minutes. We have added more information in the methods section and the time of exposure in the legend of the figure 2.

Line 528-529 : “Pictures were taken using a Nikon Eclipse Ts2R microscope (500 ms of exposure) [...]”.

Titers of each virus used should be disclosed in the Methods section.

We have added a table in the methods section with the titers of all the viruses used in this study. The titrations of the viral stocks of SARS-CoV-2 that we used here have been repeated several times as a control for other projects and the titers have indeed been confirmed.

Line 643 : “Table 1: Titer of our viral stocks”.

Is the SARS-CoV-2 infection pfu of 10,000 accurate in ED Fig. 1?

In order to infect with 10000pfu we have added 2 μ l drops of stocks (10⁶pfu/ml) on the top of the slice. After the slice has absorbed the drop we have repeated until 10 μ l per slice (5 times 2 μ l). Our results using RT-qPCR quantification at day 0 show very low variability among infected slices, highlighting the accuracy of the infection.

Peer Review, second round –

Reviewer #1 (Remarks to the Author):

The authors have adequately addressed the comments of the reviewers.

Reviewer #2 (Remarks to the Author):

The authors of this manuscript have responded very clearly to all of the authors comments. There remains an important point which speaks to the question of importance of the paper, which is related to its relevance to SARS-CoV-2 infection in human brain. The response to reviewer 3, the authors state that the main idea behind the study was to determine what would be the consequences if the virus reached brain parenchyma. This seems at odds with the response to reviewer 1, where the reviewer asks how the virus is thought to get across to the brain from the respiratory tract. The response suggests that irrespective of the route of infection, the model "establishes" the hamster brainstem as a model to address the SARS-CoV2CNS pathogenesis. This statement seems an over-reach, as the establishment of a relevant model would need to establish its relevance to human CNS disease. The authors further suggest the route of infection as well as references showing virus in the brains of patients in line 52-58.

The references selected here are valid, however, the overwhelming evidence suggest that this virus does not actually infect neurons, and that vascular endothelial cells represent areas of infection and disease, providing an alternative route to the brain parenchyma. The role of inflammation in disease is likely highly important as the direct infection seems unlikely.

Furthermore, the discussion in lines 52-58 is very strong in favor of the neuronal infection model, despite the preponderance of evidence that the virus does not infect neurons. While a citation is given for virus in CSF, an overwhelming majority of papers suggest that virus is not detected in CSF. If the discussion was more balanced, the manuscript might not have such significant impact. The idea that the organotypic cultures represent a useful model to screen antivirals is dependent on the interpretation of this study in the overall context of what is known. The infection of organs outside the lungs may also represent late rather than early events in the viral life cycle. Antivirals also do not appear to be effective during the late events in course of disease.

Reviewer #3 (Remarks to the Author):

The authors have addressed my concerns to the best of their ability. The additional information and references of human subjects reports helps with the interpretation and relevance of the reported findings. I have no additional concerns.

Response to reviewers:

We acknowledge reviewer 1 and 3 for their approval

Reviewer 2 :

The authors of this manuscript have responded very clearly to all of the authors' comments.

→ We thank Reviewer 2 for this encouraging comment

There remains an important point which speaks to the question of importance of the paper, which is related to its relevance to SARS-CoV-2 infection in human brain. The response to reviewer 3, the authors state that the main idea behind the study was to determine what would be the consequences if the virus reached brain parenchyma. This seems at odds with the response to reviewer 1, where the reviewer asks how the virus is thought to get across to the brain from the respiratory tract. The response suggests that irrespective of the route of infection, the model "establishes" the hamster brainstem as a model to address the SARS-CoV-2 CNS pathogenesis. This statement seems an over-reach, as the establishment of a relevant model would need to establish its relevance to human CNS disease. The authors further suggest the route of infection as well as references showing virus in the brains of patients in line 52-58.

→ We have added several sentences to attenuate the statement of using brainstem slices as model to address SARS-CoV-2 CNS pathogenesis which is visibly broader than initially thought. We have also specified at multiple places that these models are not dedicated to investigate how the virus reaches the CNS and we have replaced pathogenesis with infection.

The references selected here are valid, however, the overwhelming evidence suggest that this virus does not actually infect neurons, and that vascular endothelial cells represent areas of infection and disease, providing an alternative route to the brain parenchyma. The role of inflammation in disease is likely highly important as the direct infection seems unlikely. Furthermore, the discussion in lines 52-58 is very strong in favor of the neuronal infection model, despite the preponderance of evidence that the virus does not infect neurons. While a citation is given for virus in CSF, an overwhelming majority of papers suggest that virus is not detected in CSF. If the discussion was more balanced, the manuscript might not have such significant impact.

→ We agree that SARS-CoV-2 is not as neurotropic as Nipah virus. Here the idea was to show that if the virus reaches the brain parenchyma, then it can infect only (and mainly) specific classes of neurons (i.e. granular and Golgi neurons) and some specific area notably the highly suspected vagus nerve region in the Brainstem. The highlight of different classes of neurons in periphery is now established (olfactory, lungs, gut) as well as the in vitro susceptibility in cells and in human brain organoids. We have thus tempered the importance of neuronal infection without excluding it since more and more publications also demonstrates that virus can be found in CSF when using appropriate methods and infect neurons in a subset of patient. We have also added numerous case report confirming the detection of viral RNA in CSF in a subset of patients. An explanation would be that as for numerous other viruses the post mortem observation might be too late with viruses killing cells they infect. An alternative would be that the virus only reaches the CNS when the cytokine storm is not triggered thus only in patients who are not hospitalized or not dying. Finally the low level of virus found in CSF may be due to a low efficacy of the virus to bud in the CNS as observed for Measles virus. In any case this will require further investigation before make conclusions.

The idea that the organotypic cultures represent a useful model to screen antivirals is dependent on the interpretation of this study in the overall context of what is known. The infection of organs outside the lungs may also represent late rather than early events in the viral life cycle. Antivirals also do not appear to be effective during the late events in course of disease.

→ Of course we totally agree with reviewer 2 that any really efficient treatment will only work in the early phases of infection with any cytopathic virus. Once the lungs are destroyed by the infection there is unfortunately no treatment to completely regenerate the organ and save the patient. Here the question is not relying on the interpretation since drugs are tested in the lungs in the early phases of the infection. If they already do not work or if they are toxic then it does not make sense to go further in *in vivo* studies. The models allow better appreciating antiviral efficacy in the organic context as well as the direct tissue toxicity of the compounds and reducing the animal payload in the meantime. The toxicity cannot only be addressed in lungs for compounds which can enter the blood circulation (either naturally or because of barrier disruption). Thus, we have specified that the models allow testing antiviral drugs potentially efficient in the early stages of the infection.